# OpenXAI: Towards a Transparent Evaluation of Post hoc Model Explanations

Chirag Agarwal[1,2], Satyapriya Krishna[1], Eshika Saxena[1], Martin Pawelczyk[3], Nari Johnson[4], Isha Puri[1], Marinka Zitnik[1], and Himabindu Lakkaraju[1]

[1]Harvard University
[2]Adobe
[3]University of Tubingen
[4]Carnegie Mellon University

## Abstract

While several types of post hoc explanation methods have been proposed in recent literature, there is very little work on systematically benchmarking these methods. Here, we introduce OpenXAI, a comprehensive and extensible open-source framework for evaluating and benchmarking post hoc explanation methods. OpenXAI comprises of the following key components: (i) a flexible synthetic data generator and a collection of diverse real-world datasets, pre-trained models, and state-of-the-art feature attribution methods, (ii) open-source implementations of twenty-two quantitative metrics for evaluating faithfulness, stability (robustness), and fairness of explanation methods, and (iii) the first ever public XAI leaderboards to readily compare several explanation methods across a wide variety of metrics, models, and datasets. OpenXAI is easily extensible, as users can readily evaluate custom explanation methods and incorporate them into our leaderboards.

Overall, OpenXAI provides an automated end-to-end pipeline that not only simplifies and standardizes the evaluation of post hoc explanation methods, but also promotes transparency and reproducibility in benchmarking these methods. While the first release of OpenXAI supports only tabular datasets, the explanation methods and metrics that we consider are general enough to be applicable to other data modalities. OpenXAI datasets and data loaders, implementations of state-of-the-art explanation methods and evaluation metrics, as well as leaderboards are publicly available at `https://open-xai.github.io/`. OpenXAI will be regularly updated to incorporate text and image datasets, other new metrics and explanation methods, and welcomes inputs from the community.

## 1 Introduction

As predictive models are increasingly deployed in critical domains (e.g., healthcare, law, and finance), there has been a growing emphasis on explaining the predictions of these models to decision makers (e.g. doctors, judges) so that they can understand the rationale behind model predictions, and determine if and when to rely on these predictions. To this end, various techniques have been proposed in recent literature to generate post hoc explanations of individual predictions made by complex ML models. Several of such *local explanation methods* output the influence of each of the features on the model's prediction, and are therefore referred to as *local feature attribution methods*. Due to their generality, feature attribution methods are increasingly being utilized to explain complex models in medicine, finance, law, and science [23, 34, 78]. Thus, it is critical to ensure that the explanations generated by these methods are *reliable* so that relevant stakeholders and decision makers are provided with credible information about the underlying models [6].

36th Conference on Neural Information Processing Systems (NeurIPS 2022) Track on Datasets and Benchmarks.

Prior works have studied several notions of explanation reliability such as *faithfulness* (or fidelity) [81, 51, 33], *stability* (or robustness) [7, 4], and *fairness* [18, 9], and proposed metrics for quantifying these notions. Many of these works also demonstrated through small-scale experiments or qualitative analysis that certain explanation methods are not effective w.r.t. specific notions of reliability. For instance, Alvarez-Melis and Jaakkola [7] visualized the explanations generated by some of the popular gradient based explanation methods [67, 66, 70, 72] for MNIST images, and showed that they are not robust to small input perturbations. However, it is unclear if such findings generalize beyond the settings studied. More broadly, one of the biggest open questions which has far-reaching implications for the progress of explainable AI (XAI) research is: *which explanation methods* are effective w.r.t. *what notions of reliability* and *under what conditions*? [43]. A first step towards answering this question involves systematically benchmarking explanation methods in a reproducible and transparent manner. However, the increasing diversity of explanation methods, and the plethora of evaluation settings and metrics outlined in existing research without standardized open-source implementations make it rather challenging to carry out such benchmarking efforts.

In this work, we address the aforementioned challenges by introducing OpenXAI, a comprehensive and extensible open-source framework for systematically and efficiently benchmarking explanation methods in a transparent and reproducible fashion. More specifically, our work makes the following key contributions:

1. We introduce the OpenXAI framework, an *open-source ecosystem* designed to support systematic, reproducible, and efficient evaluations of post hoc explanation methods. OpenXAI unifies the existing scattered repositories of datasets, models, and evaluation metrics, and provides a *simple and easy-to-use API* that enables researchers and practitioners to benchmark explanation methods using just a few lines of code (Section 2).

2. Our OpenXAI framework currently provides open-source implementations and ready-to-use API interfaces for *seven state-of-the-art feature attribution methods* (LIME, SHAP, Vanilla Gradients, Gradient x Input, SmoothGrad, and Integrated Gradients), and *twenty-two quantitative metrics* to evaluate the faithfulness, stability, and fairness of feature attribution methods. In addition, it includes a comprehensive collection of *seven real-world datasets* spanning diverse real-world domains, and *sixteen different pre-trained models*. OpenXAI also introduces *a novel and flexible synthetic data generator* to synthesize datasets of varying sizes, complexity, and dimensionality which facilitate the construction of reliable ground truth explanations (Section 2).

3. As part of our OpenXAI framework, we also develop the first-ever *public XAI leaderboards* (shown in Figure 1) to promote transparency, and to allow users to easily compare the performance of multiple explanation methods across a wide variety of synthetic and real-world datasets, evaluation metrics, and predictive models.

4. OpenXAI framework is *easily extensible* i.e., researchers and practitioners can readily incorporate custom explanation methods, datasets, predictive models, and evaluation metrics into our framework and leaderboards (Section 2).

5. Lastly, using our proposed OpenXAI framework, we *perform rigorous empirical benchmarking* of the aforementioned state-of-the-art feature attribution methods to determine which methods are effective w.r.t. what notions of reliability across a wide variety of datasets and predictive models (Section 3).

Overall, our OpenXAI framework provides an end-to-end pipeline that unifies, simplifies, and standardizes several existing workflows to evaluate explanation methods. By enabling systematic and efficient evaluation and benchmarking of existing and new explanation methods, our OpenXAI framework can inform and accelerate new research in the emerging field of XAI. OpenXAI will be regularly updated and welcomes input from the community.

**Related Work.** Our work builds on the vast literature in explainable AI. Here, we discuss closely related works and their connections to our benchmark. A more detailed discussion of the related work is included in the Appendix.

*Evaluation Metrics for Post hoc Explanations*: Prior research has studied several notions of explanation reliability, namely, *faithfulness* (or fidelity), *stability* (or robustness), and *fairness* [51, 81, 7, 18]. While the faithfulness notion captures how faithfully a given explanation captures the true behavior of the underlying model [81, 51, 33], stability ensures that explanations do not change

drastically with small perturbations to the input [29, 7]. The fairness notion, on the other hand, ensures that there are no group-based disparities in the faithfulness or stability of explanations [18]. To this end, prior works [51, 69, 81, 7, 18, 33] proposed various evaluation metrics to quantify the aforementioned notions. For instance, Petsiuk et al. [59] measured the change in the probability of the predicted class when important features (as identified by an explanation) are deleted from or introduced into the data instance. A sharp change in the probability implies a high degree of explanation faithfulness. Alvarez-Melis and Jaakkola [7] loosely quantified stability as the maximum change in the resulting explanations when small perturbations are made to a given instance. Dai et al. [18] quantified unfairness of explanations as the difference between the faithfulness (or stability) metric values averaged over instances in the majority and the minority subgroups.

*XAI Libraries and Benchmarks:* Prior works have introduced a few XAI libraries and benchmarks, the most popular among them being Captum [42], Quantus [32], XAI-Bench [51], and SHAP Benchmark. Below, we provide a brief description of each of these, and detail how our work differs from them.

While ***Captum library*** [42] is an open-source library which provides implementations and APIs for various state-of-the-art explanation methods, its focus is not on evaluating and/or benchmarking these methods which is the main goal of our work. ***Quantus library*** [32], on the other hand, provides implementations of certain evaluation metrics to measure the faithfulness and stability/robustness of explanation methods. However, it does not focus on benchmarking explanation methods or providing public dashboards to compare the performance of these methods. Furthermore, the stability/robustness measures [7] supported by Quantus are somewhat outdated and have been superseded by recently proposed metrics [4]. In addition, Quantus does not support any fairness metrics to evaluate disparities in the quality of explanations which is very important in real-world settings such as healthcare, criminal justice, and policy. In contrast, OpenXAI not only subsumes popular faithfulness and stability/robustness metrics supported by Quantus but also supports 19 new metrics to measure the faithfulness, stability/robustness, as well as the fairness of explanation methods [4, 18, 43]. In addition, OpenXAI focuses on systematically benchmarking state-of-the-art explanation methods and providing public dashboards to readily compare these methods.

***SHAP benchmark*** [2] only focuses on evaluating and comparing different variants of SHAP [54] via certain faithfulness metrics which are similar to the Prediction Gap on Important (PGI) and Unimportant (PGU) feature perturbation metrics outlined in our work. Note that the SHAP benchmark does not include any stability/robustness or fairness metrics. In contrast, OpenXAI not only includes 20 new metrics to evaluate the stability/robustness and fairness of explanation methods but also benchmarks various other methods (e.g., LIME, Gradient-based methods).

***XAI-Bench*** [51] constructed synthetic datasets with ground truth explanations to evaluate the faithfulness of a few explanation methods (e.g., LIME, SHAP, MAPLE). However, recent research argued that their evaluation is unreliable, and predictive models learned using their synthetic datasets may not adhere to the ground truth explanations [24]. In addition, the aforementioned evaluation is rather limited in scope as synthetic datasets may not even be representative of real-world data [24]. In contrast, our work not only proposes a novel synthetic data generator that addresses the shortcomings of the synthetic datasets constructed in XAI-Bench but also facilitates the evaluation and benchmarking of the faithfulness, stability, as well as the fairness of 7 state-of-the-art explanation methods on 7 real-world datasets with no ground truth explanations.

In summary, our work is significantly different from existing libraries and benchmarks, and makes the following key contributions:

- We provide implementations and easy-to-use API interfaces for 22 metrics to evaluate the faithfulness, stability, and fairness of explanation methods. 18 out of the 22 state-of-the-art metrics included in OpenXAI have not been implemented in any prior libraries or benchmarks – e.g., faithfulness metrics such as Feature Agreement (FA), Rank Agreement (RA), Sign Agreement (SA), Signed Rank Agreement (SRA), Pairwise Rank Agreement (PRA), stability metrics such as Relative Representation Stability (RRS), Relative Output Stability (ROS), and all fairness metrics.
- We also introduce a novel and flexible synthetic data generator to synthesize datasets of varying sizes, complexity, and dimensionality to facilitate the construction of reliable ground truth explanations in order to evaluate state-of-the-art explanation methods. Our synthetic data generator addresses the shortcomings of the prior synthetic benchmark (XAI-

Bench) by generating synthetic datasets which encapsulate certain key properties, namely, unambiguously defined local neighborhoods, a clear description of feature importances in each local neighborhood, and feature independence. These properties, in turn, allow us to theoretically guarantee that any accurate model trained on our synthetic datasets will adhere to the ground truth explanations of the underlying data.

- We perform rigorous empirical benchmarking of 7 state-of-the-art feature attribution methods using our OpenXAI framework to determine which methods are effective w.r.t. each of the 22 evaluation metrics across 8 real-world and synthetic datasets, and 16 different predictive models. Note that none of the previously proposed libraries or benchmarks carry out such exhaustive benchmarking efforts across such a wide variety of metrics, models, and datasets. We also introduce the first ever public XAI leaderboards with such a wide variety of explanation methods, metrics, models, and datasets, to promote transparency and showcase the results of our benchmarking efforts.

## 2    Overview of OpenXAI Framework

OpenXAI provides a comprehensive programmatic environment with synthetic and real-world datasets, data processing functions, explainers, and evaluation metrics to rigorously and efficiently benchmark explanation methods. Below, we discuss each of these components in detail.

**1) Datasets and Predictive Models.** The current release of our OpenXAI framework includes a collection of eight different synthetic and real-world datasets. While synthetic datasets allow us to construct ground truth explanations which can then be used to evaluate explanations output by state-of-the-art methods, real-world datasets (where it is typically hard to construct ground truth explanations) help us benchmark these methods in a more realistic manner suitable for practical applications [51]. We would like to note that OpenXAI includes datasets that are widely employed in XAI research to evaluate the efficacy of newly proposed methods and study the behavior of existing methods [9, 18–20, 38, 69, 73].

*Synthetic Datasets*: While prior research [51, 41] proposed methods to generate synthetic datasets and corresponding ground truth explanations, they all suffer from a significant drawback as demonstrated by Faber et al. [24] – there is no guarantee that the models trained on these datasets will adhere to the ground truth explanations of the underlying data. This, in turn, implies that evaluating post hoc explanations using the above ground truth explanations would be incorrect since post hoc explanations are supposed to reliably explain the behavior of the underlying model, and not that of the underlying data. To illustrate, let us consider the case where we use aforementioned methods to construct a synthetic dataset with features $A, B, C$, and $D$ such that the ground truth labels only depend on features $A$ and $B$ i.e., the ground truth explanation of the underlying data indicates that features $A$ and $B$ are most important. If we train a model on this data and if features $A$ and $B$ are correlated with $C$ and $D$ respectively, then the resulting model may base its predictions on $C$ and $D$ (and not $A$ and $B$) and still be very accurate. If a post hoc explanation of this model then (correctly) indicates that the most important features of the model are $C$ and $D$, this explanation may be deemed incorrect if we compare it against the ground truth explanation of the underlying data. This problem further exacerbates as we increase the complexity of the ground truth labeling function [24].

To address the aforementioned challenges, we develop a novel synthetic data generation mechanism, *SynthGauss*, which encapsulates three key properties, namely, feature independence, unambiguously-defined local neighborhoods, and a clear description of feature influence in each local neighborhood. Intuitively, this approach generates $K$ well-separated clusters where points in each cluster $k \in \{1, 2, \cdots, K\}$ are sampled from a Gaussian distribution $\mathcal{N}(\mu_k, \Sigma_k)$ where $u_k \in R^d$ is the mean and $\Sigma_k \in R^{d \times d}$ is the covariance matrix. While this parameterization is general enough to support the construction of synthetic datasets of $K$ clusters with varying means and covariances, we set the means of all the clusters such that the intracluster distances are significantly smaller than the intercluster distances, and we set the covariance matrices of all the clusters to identity. This ensures that all the features are independent, and local neighborhoods (clusters) are unambiguously defined.

We then generate ground truth labels for instances by first randomly sampling feature mask vectors $m_k \in \{0, 1\}^d$ (vectors comprising of 0s and 1s) for each cluster $k$. The vector $m_k$ determines which features influence the ground truth labeling process for instances in cluster $k$ (a value of 1 indicates that the corresponding feature is influential). We then randomly sample feature weight

vectors $w_k \in R^d$ which capture the relative importance of each of the features in the labeling process of instances in each cluster $k$. The ground truth labels of instances in each cluster $k$ are then computed as a function (e.g., sigmoid) of the feature values of individual instances, and the dot product of the corresponding cluster's feature mask vector and weight vector i.e., $m_k \odot w_k$. Complete pseudocode and other details of this generation process are included in the Appendix. Note that $m_k$ corresponds to the ground truth explanation for all instances in cluster $k$. Since our generation process is designed to encapsulate feature independence, unambiguous definitions of local neighborhoods, and clear descriptions of feature influences, any accurate model trained on the resulting dataset will adhere to the ground truth explanations of the underlying data (See Theorem 1 in Appendix).

*Real-world Datasets*: In the current release of OpenXAI, we include seven real-world datasets that are highly diverse in terms of several key properties. They comprise of data spanning multiple real-world domains (e.g., finance, lending, healthcare, and criminal justice), varying dataset sizes (e.g., small vs. large-scale), dimensionalities (e.g., low vs. high dimensional), class imbalance ratios, and feature types (e.g., continuous vs. discrete). We focus on tabular data in this release as such data is commonly encountered in real-world applications where explainability is critical [75], and has also been widely studied in XAI literature [51]. Table 1 provides a summary of the real-world datasets currently included in OpenXAI. See Section E.1 in the Appendix for detailed descriptions of individual datasets. While these real-world datasets are primarily drawn from prior research and existing repositories, OpenXAI provides comprehensive data loading and pre-processing capabilities to make these datasets *XAI-ready* (more details below). We also plan to expand our collection of real-world datasets in the next iteration. Adding a new dataset into our collection is as simple as uploading a .csv file or a .zip folder. Users can also submit requests to incorporate new datasets into the OpenXAI framework by filling a simple form and providing links to the datasets (See Appendix).

**Table 1: Summary of currently available datasets in OpenXAI.** Here, "feature types" denotes whether features in the dataset are discrete or continuous, "feature information" describes what kind of information is captured in the dataset, and "balanced" denotes whether the dataset is balanced w.r.t. the predictive label.

| Dataset | Size | # features | Feature types | Feature information | Balanced |
|---|---|---|---|---|---|
| Synthetic Data | 5,000 | 20 | continuous | synthetic | ✓ |
| German Credit [22] | 1,000 | 20 | discrete, continuous | demographic, personal, financial | ✗ |
| HELOC [25] | 9,871 | 23 | continuous | demographic, financial | ✓ |
| COMPAS [36] | 18,876 | 7 | discrete, continuous | demographic, personal, criminal | ✗ |
| Adult Income [79] | 48,842 | 13 | discrete, continuous | demographic, personal, education/employment, financial | ✗ |
| Give Me Some Credit [27] | 102,209 | 10 | discrete, continuous | demographic, personal, financial | ✗ |
| Pima-Indians Diabetes [71] | 768 | 9 | discrete, continuous | demographic, healthcare | ✗ |
| Framingham heart study [1] | 4,240 | 16 | continuous | demographic, healthcare | ✗ |

*Data loaders and pre-trained models*: OpenXAI provides a `Dataloader` class that can be used to load the aforementioned collection of synthetic and real-world datasets as well as any other custom datasets, and ensures that they are *XAI-ready*. More specifically, this class takes as input the name of an existing OpenXAI dataset or a new dataset (name of the .csv file), and outputs a train set which can then be used to train a predictive model, a test set which can be used to generate local explanations of the trained model, as well as any ground-truth explanations (if and when available). If the dataset already comes with pre-determined train and test splits, this class loads train and test sets from those pre-determined splits. Otherwise, it divides the entire dataset randomly into train (70%) and test (30%) sets. Users can also customize the percentages of train-test splits. The code snippet below shows how to import the `Dataloader` class and load an existing OpenXAI dataset.

```
from OpenXAI import Dataloader
loader_train, loader_test = Dataloader.return_loaders(data_name='german',
download=True)
inputs, labels = iter(loader_test).next()
```

We also pre-trained two classes of predictive models (e.g., deep neural networks of varying degrees of complexity, logistic regression models etc.) and incorporated them into the OpenXAI framework so that they can be readily used for benchmarking explanation methods. The code snippet below shows how to load OpenXAI's pre-trained models using our `LoadModel` class.

```
from OpenXAI import LoadModel
model = LoadModel(data_name='german', ml_model='ann')
```

Adding additional pre-trained models into the OpenXAI framework is as simple as uploading a file with details about model architecture and parameters in a specific template. Users can also submit requests to incorporate custom pre-trained models into the OpenXAI framework by filling a simple form and providing details about model architecture and parameters (See Appendix).

**2) Explainers.** OpenXAI provides ready-to-use implementations of six state-of-the-art feature attribution methods, namely, LIME, SHAP, Vanilla Gradients, Gradient x Input, SmoothGrad, and Integrated Gradients. An implementation of a random baseline which randomly assigns importance values to each of the features, and returns these random assignments as explanations is also included. Our implementations of these methods build on other open-source libraries (e.g., Captum [42]) as well as their original implementations. While methods such as LIME and SHAP leverage perturbations of data instances and their corresponding model predictions to *learn* a local explanation model, they do not require access to the internals of the models or their gradients. On the other hand, Vanilla Gradients, Gradient x Input, SmoothGrad, and Integrated Gradients require access to the gradients of the underlying models but do not need to repeatedly query the models for their predictions (see Table 6 in Appendix for a brief summary of these methods). These differences influence the efficiency with which explanations can be generated by these methods. OpenXAI provides an abstract `Explainer` class which enables us to load existing explanation methods as well as integrate new explanation methods.

```
from OpenXAI import Explainer
exp_method = Explainer(method='LIME')
explanations = exp_method.get_explanations(model, X=inputs, y=labels)
```

All the explanation methods included in OpenXAI are readily accessible through the `Explainer` class, and users just have to specify the method name in order to invoke the appropriate method and generate explanations as shown in the above code snippet. Users can easily incorporate their own custom explanation methods into the OpenXAI framework by extending the `Explainer` class and including the code for their methods in the `get_explanations` function (see template below) of this class. They can then submit a request to incorporate their custom methods into OpenXAI library by filling a form and providing the GitHub link to their code as well as a summary of their explanation method (See Appendix).

**3) Evaluation Metrics.** OpenXAI provides implementations and ready-to-use APIs for a set of twenty-two quantitative metrics proposed by prior research to evaluate the faithfulness, stability, and fairness of explanation methods. OpenXAI is the first XAI benchmark to consider all the three aforementioned aspects of explanation reliability. More specifically, we include eight different metrics to measure explanation faithfulness (both with and without ground truth explanations) [43, 59], three different metrics to measure stability [4], and eleven different metrics to measure group-based disparities (unfairness) [18] in the values of the aforementioned faithfulness and stability metrics. The metrics that we choose are drawn from the latest works in explainable AI literature. Below, we briefly describe these metrics. Detailed descriptions of all the metrics along with notation and equations are included in the Appendix.

a) *Ground-truth Faithfulness*: Krishna et al. [43] recently proposed six evaluation metrics to capture the similarity between the top-K or a select set of features of any two feature attribution-based explanations. We leverage these metrics to capture the similarity between the explanations output by state-of-the-art methods and the ground-truth explanations constructed using our synthetic data generation process. These metrics and their definitions are given as follows: i) Feature Agreement (FA) which computes the fraction of top-K features that are common between a given post hoc explanation and the corresponding ground truth explanation, ii) Rank Agreement (RA) metric which measures the fraction of top-K features that are not only common between a given post hoc explanation and the corresponding ground truth explanation, but also have the same position in the respective rank orders, iii) Sign Agreement (SA) metric which computes the fraction of top-K

features that are not only common between a given post hoc explanation and the corresponding ground truth explanation, but also share the same sign (direction of contribution) in both the explanations, iv) Signed Rank Agreement (SRA) metric which computes the fraction of top-K features that are not only common between a given post hoc explanation and the corresponding ground truth explanation, but also share the same feature attribution sign (direction of contribution) and position (rank) in both the explanations, v) Rank Correlation (RC) metric which computes Spearman's rank correlation coefficient to measure the agreement between feature rankings provided by a given post hoc explanation and the corresponding ground truth explanation, and vi) Pairwise Rank Agreement (PRA) metric which captures if the relative ordering of every pair of features is the same for a given post hoc explanation as well as the corresponding ground-truth explanation.

b) *Predictive Faithfulness*: We leverage the metrics outlined by [59, 18] to measure the faithfulness of an explanation when no ground truth is available. This metric, referred to as Prediction Gap on Important feature perturbation (PGI), computes the difference in prediction probability that results from perturbing the features deemed as influential by a given post hoc explanation. Higher values on this metric imply greater explanation faithfulness. We also consider the converse of this metric, Prediction Gap on Unimportant feature perturbation (PGU), which perturbs the unimportant features and measures the change in prediction probability.

c) *Stability*: We consider the metrics introduced by Alvarez-Melis and Jaakkola [7], Agarwal et al. [4] to measure how robust a given explanation is to small input perturbations. More specifically, we leverage the metrics Relative Input Stability (RIS), Relative Representation Stability (RRS), and Relative Output Stability (ROS) which measure the maximum change in explanation relative to changes in the inputs, model parameters, and output prediction probabilities respectively.

d) *Fairness*: Following the work by Dai et al. [18], we measure the fairness of post hoc explanations by averaging all the aforementioned metric values across instances in the majority and minority subgroups, and comparing the two estimates. If there is a huge difference in the two estimates, then we consider this to be evidence for unfairness.

Invoking the aforementioned metrics to benchmark an explanation methods is quite simple and the code snippet below describes how to invoke the RIS metric. Users can easily incorporate their own custom evaluation metrics into OpenXAI by filling a form and providing the GitHub link to their code as well as a summary of their metric (See Appendix).

```
from OpenXAI import Evaluator
metric_evaluator = Evaluator(inputs, labels, model, explanations)
score = metric_evaluator.eval(metric='RIS')
```

***Benchmarking***: As can be seen from the code snippets in this section, OpenXAI allows end users to easily benchmark explanation methods using just a few lines of code. To summarize the benchmarking process, let us consider a scenario where we would like to benchmark a new explanation method using OpenXAI's pre-trained neural network model and the German Credit dataset. First, we use OpenXAI's `Dataloader` class to load the German Credit dataset. Second, we load the neural network model ('ann') using our `LoadModel` class. Third, we extend the `Explainer` class and incorporate the code for the new explanation method in the `get_explanation` function of this class. Finally, we evaluate the new explanation method using various metrics from the `Evaluator` class.

**4) Leaderboards.** OpenXAI introduces the first ever public XAI leaderboards to promote transparency, and enable users to easily compare the performance of multiple explanation methods across a variety of evaluation metrics, predictive models, and datasets. In the current release, we have six different leaderboards each corresponding to a particular dataset. A snapshot of one of our leaderboard pages is shown in Figure 1. Users can submit requests for their custom explanation methods to be featured on one of our leaderboards. To this end, they first need to following the aforementioned *benchmarking process* to develop and evaluate their explanation method.

## 3   Benchmarking Analysis

Next, we describe how we benchmark state-of-the-art explanation methods using our OpenXAI framework, and also discuss key findings of this benchmarking analysis. Code to reproduce all the results is available at https://github.com/AI4LIFE-GROUP/OpenXAI.

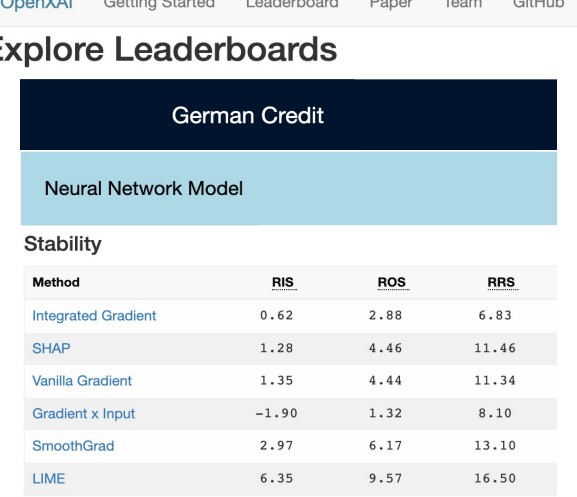

**Figure 1: A snapshot of the leaderboard page from OpenXAI public website.** We also provide interactive ranking functionality (arrow mark in the figure) which allows users to rank explanation methods based on metrics of their choice. Please visit the website to see leaderboards for other datasets.

**Experimental Setup.** We benchmark all the six state-of-the-art feature attribution methods currently available in our OpenXAI framework along with the random baseline, using the `openxai.Evaluator` module (See Section 2). We use default hyperparameter settings for all these methods following the guidelines outlined in the original implementations. Details about the hyperparameters used in our experiments are discussed in Section E.3 in the Appendix. Our OpenXAI framework currently has two pre-trained models, a logistic regression model and a deep neural network model, for each dataset. The neural network models have two fully connected hidden layers with 100 nodes in each layer, and they use ReLU activation functions and an output softmax layer. See Appendix E.4 for more details on model architectures, model training, and model performance.

**Table 2: Ground-truth and predicted faithfulness results on the Heloc dataset for all explanation methods with LR model.** Shown are average and standard error metric values computed across all instances in the test set. ↑ indicates that higher values are better, and ↓ indicates that lower values are better. Values corresponding to best performance are bolded.

| Method | PRA (↑) | RC (↑) | FA (↑) | RA (↑) | SA (↑) | SRA (↑) | PGU (↓) | PGI (↑) |
|---|---|---|---|---|---|---|---|---|
| Random | 0.500±0.00 | 0.005±0.01 | 0.498±0.00 | 0.043±0.00 | 0.251±0.00 | 0.022±0.00 | 0.033±0.00 | 0.035±0.00 |
| VanillaGrad | **1.0**±0.00 | **1.0**±0.00 | **0.957**±0.00 | **0.957**±0.00 | 0.469±0.01 | 0.469±0.01 | 0.034±0.00 | 0.036±0.00 |
| IntegratedGrad | 1.0±0.00 | 1.0±0.00 | 0.957±0.00 | 0.957±0.00 | 0.469±0.01 | 0.469±0.01 | 0.034±0.00 | 0.036±0.00 |
| Gradient x Input | 0.641±0.00 | 0.390±0.01 | 0.582±0.00 | 0.049±0.00 | 0.255±0.01 | 0.022±0.00 | 0.033±0.00 | 0.036±0.00 |
| SmoothGrad | 1.0±0.00 | 1.0±0.00 | 0.957±0.00 | 0.957±0.00 | 0.274±0.00 | 0.274±0.00 | **0.015**±0.00 | **0.047**±0.00 |
| SHAP | 0.645±0.00 | 0.384±0.01 | 0.586±0.00 | 0.054±0.00 | 0.269±0.01 | 0.024±0.00 | 0.033±0.00 | 0.036±0.00 |
| LIME | 0.982±0.00 | 0.994±0.00 | 0.932±0.00 | 0.671±0.00 | **0.929**±0.00 | **0.670**±0.00 | 0.042±0.00 | 0.029±0.001 |

**Faithfulness.** We evaluate the ground-truth and predictive faithfulness of explanations generated by state-of-the-art methods using both synthetic and real-world datasets.

*Ground-truth faithfulness*: We evaluate ground-truth faithfulness by calculating the similarity between the generated explanations and the ground-truth explanations using the metrics discussed in Section 2. Results for various ground-truth faithfulness metrics are shown in Tables 2, 3, 16, 17. Vanilla Gradients, SmoothGrad, and Integrated Gradients produce explanations that achieve perfect scores on four ground-truth faithfulness metrics, viz. pairwise rank agreement (PRA), feature agreement (FA), rank agreement (RA), and rank correlation (RC) metrics, for all datasets. However, on average, across all datasets, LIME outperforms other methods on the signed agreement (SA) [+61.6%] and signed-rank agreement (SRA) [+65.3%] metrics, whereas gradient-based explainers achieve relatively lower values. While illustrative in nature, these findings show how OpenXAI can help identify the limitations of existing explanation methods, which in turn can inform the design of new methods.

**Table 3: Ground-truth and predicted faithfulness results on the Adult Income dataset for all explanation methods with LR model.** Shown are average and standard error metric values computed across all instances in the test set. ↑ indicates that higher values are better, and ↓ indicates that lower values are better. Values corresponding to best performance are bolded.

| Method | PRA (↑) | RC (↑) | FA (↑) | RA (↑) | SA (↑) | SRA (↑) | PGU (↓) | PGI (↑) |
|---|---|---|---|---|---|---|---|---|
| Random | 0.499±0.00 | 0.0±0.00 | 0.496±0.00 | 0.068±0.00 | 0.250±0.00 | 0.037±0.00 | 0.053±0.00 | 0.06±0.00 |
| VanillaGrad | 1.±0.00 | 1.±0.00 | 0.923±0.00 | 0.921±0.00 | 0.138±0.00 | 0.136±0.00 | 0.07±0.001 | 0.039±0.001 |
| IntegratedGrad | 1.±0.00 | 1.±0.00 | **0.923**±0.00 | 0.923±0.00 | 0.138±0.00 | 0.138±0.00 | 0.07±0.001 | 0.039±0.001 |
| Gradient x Input | 0.580±0.00 | 0.281±0.00 | 0.567±0.00 | 0.075±0.00 | 0.070±0.00 | 0.003±0.00 | 0.043±0.00 | 0.073±0.00 |
| SmoothGrad | **1.**±0.00 | **1.**±0.00 | 0.923±0.00 | **0.923**±0.00 | 0.741±0.00 | **0.741**±0.00 | **0.008**±0.00 | **0.099**±0.001 |
| SHAP | 0.655±0.00 | 0.379±0.00 | 0.601±0.00 | 0.105±0.00 | 0.133±0.00 | 0.009±0.00 | 0.047±0.00 | 0.068±0.00 |
| LIME | 0.913±0.00 | 0.921±0.00 | 0.869±0.00 | 0.697±0.00 | **0.858**±0.00 | 0.689±0.00 | 0.014±0.00 | 0.094±0.001 |

*Predictive faithfulness*: Tables 2, 3, 16, 17 show results for the PGI and PGU metrics implemented in OpenXAI (see Section 2 and Appendix A). Overall, we find that SmoothGrad explanations are most faithful to the underlying model and, on average, across multiple datasets outperform other feature-attribution methods on PGU metric (+43.03%). However, results from the German credit dataset for the ANN model show that Gradient x Input produces considerably more faithful (+6.74%) explanations than other methods. Finally, this analysis confirms the finding by Krishna et al. [43] that explanations output by state-of-the-art methods do not necessarily align with each other. This finding further highlights the need for rigorous empirical and theoretical benchmarking of explanation methods.

**Stability.** Next, we examine the stability of explanation methods when the underlying models are LR models in Tables 4 and 5, and neural network models in Tables 19 and 20 in the Appendix. Due to space constraints, we focus on RIS and RRS metrics in the main paper and leave the other results to the Appendix. Overall, the relative stability varies considerably across different datasets, implying that no single explanation method is consistently the most stable. First, for the synthetic dataset in Table 4, we find that Gradient x Input, on average, outperforms feature-attribution methods in relative input stability (+93.5%, RIS) and relative representation stability (+59.2%, RRS). However, stability of Gradient x Input significantly degrades on real-world datasets (Table 5, 19, 20). Second, as shown in Tables 5, 19, and 20, there is no single explanation method that has the highest input and representation stability across all the real-world datasets. On average, across all real-world datasets, SmoothGrad achieves 63.2% higher RRS values compared to other methods, whereas no method performs consistently well when it comes to the RIS metric.

**Table 4: Stability of explanation methods on the Synthetic dataset with LR model.** Shown are average and standard error values across all test set instances. Values closer to zero are desirable, and the best performance is bolded.

| Method | RIS | RRS |
|---|---|---|
| Random | 6.868±0.013 | 6.687±0.015 |
| Vanilla Gradients | 6.133±0.011 | 6.144±0.006 |
| Integrated Gradients | 5.957±0.013 | 9.022±0.043 |
| Gradient x Input | **0.405**±0.015 | **3.422**±0.037 |
| SmoothGrad | 5.249±0.008 | 9.419±0.037 |
| SHAP | 5.673±0.012 | 8.751±0.035 |
| LIME | 9.355±0.008 | 13.564±0.036 |

**Table 5: Stability of explanation methods on the German Credit dataset with LR model.** Shown are average and standard error values across all test set instances. Values closer to zero are desirable, and the best performance is bolded.

| Method | RIS | RRS |
|---|---|---|
| Random | 6.274±0.104 | 16.448±0.124 |
| Vanilla Gradients | -1.384±0.112 | 5.241±0.024 |
| Integrated Gradients | -2.004±0.119 | **4.560**±0.029 |
| Gradient x Input | -0.906±0.104 | 9.437±0.124 |
| SmoothGrad | -4.780±0.117 | 4.931±0.122 |
| SHAP | **-0.230**±0.109 | 10.056±0.115 |
| LIME | -0.698±0.109 | 9.397±0.119 |

**Fairness.** To measure fairness of explanation methods, we compute the average metric values (for each of the aforementioned faithfulness and stability metrics) for different subgroups (e.g., male and female) in the dataset and compare them. Larger gaps between the metric values for different subgroups indicates higher disparities (unfairness). Without loss of generality, we present results using the PGU (see Section 2 and Appendix A) metric. Results for LR models in Figures 2 and 3 provide two key findings. First, the fairness analysis in Figures 2 and 3 shows that there are disparities in the faithfulness of explanations (see Section 2) output by several methods (Vanilla Gradients, Integrated Gradients, and SmoothGrad). Second, Gradient x Input results in the least amount of disparity across both the datasets. These results also suggest a trade-off between various evaluation metrics. For instance, Gradient x Input method underperforms on faithfulness and stability

metrics, but outperforms other methods (+8.9%) when it comes to fairness metrics. Given such trade-offs, practitioners can leverage the OpenXAI leaderboards (Figure 1) to select an explanation method that best meets application-specific needs. Results with NN models and other fairness metrics are included in the Appendix E.7.

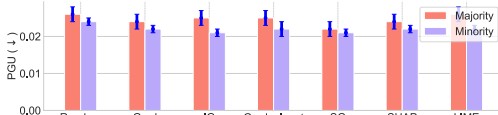 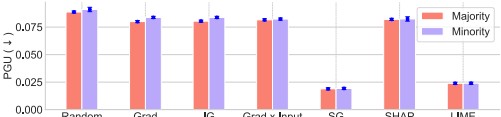

| Figure 2: **Fairness analysis of PGU metric on the German Credit dataset with LR model.** Shown are average and standard error values for majority (male) and minority (female) subgroups. Larger gaps between the values of majority and minority subgroups (i.e, red and blue bars respectively) indicate higher disparities which are undesirable. | Figure 3: **Fairness analysis of PGU metric on the Adult Income dataset with LR model.** Shown are average and standard error values for majority (male) and minority (female) subgroups. Larger gaps between the values of majority and minority subgroups (i.e, red and blue bars respectively) indicate higher disparities which are undesirable. |
|---|---|

## 4   Conclusions

As post hoc explanations are increasingly being employed to aid decision makers and relevant stakeholders in various high-stakes applications, it becomes important to ensure that these explanations are reliable. To this end, we introduce OpenXAI, an open-source ecosystem comprising of XAI-ready datasets, implementations of state-of-the-art explanation methods, evaluation metrics, leaderboards and documentation to promote transparency and collaboration around evaluations of post hoc explanations. OpenXAI can readily be used to benchmark new explanation methods as well as incorporate them into our framework and leaderboards. By enabling systematic and efficient evaluation and benchmarking of existing and new explanation methods, OpenXAI can inform and accelerate new research in the emerging field of XAI. OpenXAI will be regularly updated with new datasets, explanation methods, and evaluation metrics, and welcomes input from the community.

### Acknowledgments and Disclosure of Funding

The authors would like to thank the anonymous reviewers for their helpful feedback and all the funding agencies listed below for supporting this work. This work is supported in part by the NSF awards #IIS-2008461 and #IIS-2040989, and research awards from Google, JP Morgan, Amazon, Harvard Data Science Initiative, and $D^3$ Institute at Harvard. HL would like to thank Sujatha and Mohan Lakkaraju for their continued support and encouragement. The views expressed here are those of the authors and do not reflect the official policy or position of the funding agencies.

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
