# A Evaluation Metrics

Here, we describe different evaluation metrics that we included in the first iteration of OpenXAI. More specifically, we discuss various metrics (and their implementations) for evaluating the faithfulness, stability, and fairness of explanations generated using a given feature attribution method.

**1) Faithfulness.** To measure how faithfully a given explanation mimics the underlying model, prior work has either leveraged synthetic datasets to obtain ground truth explanations or measured the differences in predictions when feature values are perturbed [59]. Here, we discuss the two broad categories of faithfulness metrics included in OpenXAI, namely ground-truth and predictive faithfulness.

*a) Ground-truth Faithfulness.* OpenXAI leverages the following metrics outlined by Krishna et al. [43] to calculate the agreement between ground-truth explanations (i.e., coefficients of logistic regression models) and explanations generated by state-of-the-art methods.

- *Feature Agreement (FA)* metric computes the fraction of top-$K$ features that are common between a given post hoc explanation and the corresponding ground truth explanation.

- *Rank Agreement (RA)* metric measures the fraction of top-$K$ features that are not only common between a given post hoc explanation and the corresponding ground truth explanation, but also have the same position in the respective rank orders.

- *Sign Agreement (SA)* metric computes the fraction of top-$K$ features that are not only common between a given post hoc explanation and the corresponding ground truth explanation, but also share the same sign (direction of contribution) in both the explanations.

- *Signed Rank Agreement (SRA)* metric computes the fraction of top-$K$ features that are not only common between a given post hoc explanation and the corresponding ground truth explanation, but also share the same feature attribution sign (direction of contribution) and position (rank) in both the explanations.

- *Rank Correlation (RC)* metric computes the Spearman's rank correlation coefficient to measure the agreement between feature rankings provided by a given post hoc explanation and the corresponding ground truth explanation.

- *Pairwise Rank Agreement (PRA)* metric captures if the relative ordering of every pair of features is the same for a given post hoc explanation as well as the corresponding ground truth explanation i.e., if feature A is more important than B according to one explanation, then the same should be true for the other explanation. More specifically, this metric computes the fraction of feature pairs for which the relative ordering is the same between the two explanations.

**Reported metric values.** While the aforementioned metrics quantify the ground-truth faithfulness of individual explanations, we report a single value corresponding to each explanation method and dataset to facilitate easy comparison of state-of-the-art methods. To this end, we adopt a similar strategy as that of Krishna et al. [43] and compute the aforementioned metrics for each instance in the test data and then average these values. To arrive at the reported values of FA, RA, SA, and SRA metrics, instead of setting a specific value for $K$, we do the above computation for all possible values of $K$ and then plot the different values of $K$ (on the x-axis) and the corresponding averaged metric values (on the y-axis), and calculate the area under the resulting curve (AUC). On the other hand, in case of RC and PRA metrics, we set $K$ to the total number of features and compute the averaged metric values (across all test instances) as discussed above.

*b) Predictive Faithfulness.* Following Petsiuk et al. [59], OpenXAI includes two complementary predictive faithfulness metrics: i) Prediction Gap on Important feature perturbation (PGI) which measures the difference in prediction probability that results from perturbing the features deemed as influential by a given post hoc explanation, and ii) Prediction Gap on Unimportant feature perturbation (PGU) which measures the difference in prediction probability that results from perturbing the features deemed as unimportant by a given post hoc explanation.

For a given instance $\mathbf{x}$, we first obtain the prediction probability $\hat{y}$ output by the underlying model $f$, *i.e.,* $\hat{y} = f(\mathbf{x})$. Let $e_{\mathbf{x}}$ be an explanation for the model prediction of $\mathbf{x}$. In case of PGU, we then generate a perturbed instance $\mathbf{x}'$ in the local neighborhood of $\mathbf{x}$ by holding the top-$k$ features constant, and slightly perturbing the values of all the other features by adding a small amount of Gaussian

noise. In case of PGI, we generate a perturbed instance $\mathbf{x}'$ in the local neighborhood of $\mathbf{x}$ by slightly perturbing the values of the top-$k$ features by adding a small amount of Gaussian noise, and holding all the other features constant. Finally, we compute the expected value of the prediction difference between the original and perturbed instances as:

$$\text{PGI}(\mathbf{x}, f, e_{\mathbf{x}}, k) = \mathbb{E}_{\mathbf{x}' \sim \text{perturb}(\mathbf{x}, e_{\mathbf{x}}, \text{top-}K)}[|\hat{y} - f(\mathbf{x}')|], \tag{1}$$

$$\text{PGU}(\mathbf{x}, f, e_{\mathbf{x}}, k) = \mathbb{E}_{\mathbf{x}' \sim \text{perturb}(\mathbf{x}, e_{\mathbf{x}}, \text{non top-}K)}[|\hat{y} - f(\mathbf{x}')|], \tag{2}$$

where $\text{perturb}(\cdot)$ returns the noisy versions of $\mathbf{x}$ as described above.

**Reported metric values.** Similar to the ground-truth faithfulness metrics, we report PGI and PGU by calculating the AUC over all values of $K$.

**2) Stability.** We leverage the metrics Relative Input Stability (RIS), Relative Representation Stability (RRS), and Relative Output Stability (ROS) [4] which measure the maximum change in explanation relative to changes in the inputs, internal representations learned by the model, and output prediction probabilities respectively. These metrics can be written formally as:

$$\text{RIS}(\mathbf{x}, \mathbf{x}', e_{\mathbf{x}}, e_{\mathbf{x}'}) = \max_{\mathbf{x}'} \frac{|| \frac{(e_{\mathbf{x}} - e_{\mathbf{x}'})}{e_{\mathbf{x}}} ||_p}{\max(|| \frac{(\mathbf{x} - \mathbf{x}')}{\mathbf{x}} ||_p, \epsilon_{min})}, \ \forall \mathbf{x}' \text{ s.t. } \mathbf{x}' \in \mathcal{N}_{\mathbf{x}}; \ \hat{y}_{\mathbf{x}} = \hat{y}_{\mathbf{x}'} \tag{3}$$

where $\mathcal{N}_{\mathbf{x}}$ is a neighborhood of instances $\mathbf{x}'$ around $\mathbf{x}$, and $e_{\mathbf{x}}$ and $e_{\mathbf{x}'}$ denote the explanations corresponding to instances $\mathbf{x}$ and $\mathbf{x}'$, respectively. The numerator of the metric measures the $l_p$ norm of the percent change of explanation $e_{\mathbf{x}'}$ on the perturbed instance $\mathbf{x}'$ with respect to the explanation $e_{\mathbf{x}}$ on the original point $\mathbf{x}$. The denominator measures the $l_p$ norm between the (normalized) inputs $\mathbf{x}$ and $\mathbf{x}'$, and the max term in the denominator prevents division by zero.

$$\text{RRS}(\mathbf{x}, \mathbf{x}', e_{\mathbf{x}}, e_{\mathbf{x}'}) = \max_{\mathbf{x}'} \frac{|| \frac{(e_{\mathbf{x}} - e_{\mathbf{x}'})}{e_{\mathbf{x}}} ||_p}{\max(|| \frac{(\mathcal{L}_{\mathbf{x}} - \mathcal{L}_{\mathbf{x}'})}{\mathcal{L}_{\mathbf{x}}} ||_p, \epsilon_{min})}, \ \forall \mathbf{x}' \text{ s.t. } \mathbf{x}' \in \mathcal{N}_{\mathbf{x}}; \ \hat{y}_{\mathbf{x}} = \hat{y}_{\mathbf{x}'} \tag{4}$$

where $\mathcal{L}(\cdot)$ denotes the internal representations learned by the model. Without loss of generality, we use a two layer neural network model in our experiments and use the output of the first layer for computing RRS. Similarly, we can also define ROS as:

$$\text{ROS}(\mathbf{x}, \mathbf{x}', e_{\mathbf{x}}, e_{\mathbf{x}'}) = \max_{\mathbf{x}'} \frac{|| \frac{(e_{\mathbf{x}} - e_{\mathbf{x}'})}{e_{\mathbf{x}}} ||_p}{\max(|| \frac{(f(\mathbf{x}) - f(\mathbf{x}'))}{f(\mathbf{x})} ||_p, \epsilon_{min})}, \ \forall \mathbf{x}' \text{ s.t. } \mathbf{x}' \in \mathcal{N}_{\mathbf{x}}; \ \hat{y}_{\mathbf{x}} = \hat{y}_{\mathbf{x}'}, \tag{5}$$

where $f(\mathbf{x})$ and $f(\mathbf{x}')$ are the output prediction probabilities for $\mathbf{x}$ and $\mathbf{x}'$, respectively. To the best of our knowledge, OpenXAI is the first benchmark to incorporate all the above stability metrics.

**Reported metric values..** We compute the aforementioned metrics for each instance in the test set, and then average these values.

**3) Fairness.** It is important to ensure that there are no significant disparities between the reliability of post hoc explanations corresponding to instances in the majority and the minority subgroups. To this end, we average all the aforementioned metric values across instances in the majority and minority subgroups, and then compare the two estimates (See Figures 3, 2, 5 etc.) to check if there are significant disparities [18].

## B  Additional Related Work

Our work builds on the vast literature on model interpretability and explainability. Below is an overview of additional works that were not included in Section 1 due to space constraints.

**Inherently Interpretable Models and Post hoc Explanations.** Many approaches learn inherently interpretable models such as rule lists [80, 77], decision trees and decision lists [48], and others [46, 13, 53, 15]. However, complex models such as deep neural networks often achieve higher

accuracy than simpler models [62]. Thus, there has been significant interest in constructing post hoc explanations to understand their behavior. To this end, several techniques have been proposed in recent literature to construct *post hoc explanations* of complex decision models. For instance, LIME, SHAP, Anchors, BayesLIME, and BayesSHAP [62, 55, 63, 69] are considered *perturbation-based local* explanation methods because they leverage perturbations of individual instances to construct interpretable local approximations (e.g., linear models). On the other hand, methods such as Vanilla Gradients, Gradient x Input, SmoothGrad, Integrated Gradients, and GradCAM [67, 72, 65, 70] are referred to as *gradient-based local* explanation methods since they leverage gradients computed with respect to input features of individual instances to explain individual model predictions.

There has also been recent work on constructing *counterfactual explanations* which capture what changes need to be made to a given instance in order to flip its prediction [76, 74, 37, 61, 52, 11, 38, 57, 39, 58]. Such explanations can be leveraged to provide recourse to individuals negatively impacted by algorithmic decisions. An alternate class of methods referred to as *global* explanation methods attempt to summarize the behavior of black-box models as a whole rather than in relation to individual data points [47, 12]. A more detailed treatment of this topic is provided in other comprehensive survey articles [8, 30, 56, 49, 17].

In this work, we focus primarily on *local feature attribution-based post hoc explanation methods* i.e., explanation methods which attempt to explain individual model predictions by outputting a vector of feature importances. More specifically, the goal of this work is to enable systematic benchmarking of these methods in an efficient and transparent manner.

**Evaluating Post hoc Explanations.** In addition to the quantitative metrics designed to evaluate the *reliability* of post hoc explanation methods [51, 81, 7, 18] (See Section 1), prior works have also introduced *human-grounded* approaches to evaluate the *interpretability* of explanations generated by these methods [21]. For example, Lakkaraju and Bastani [45] carry out a user study to understand if misleading explanations can fool domain experts into deploying racially biased models, while Kaur et al. [40] find that explanations are often over-trusted and misused. Similarly, Poursabzi-Sangdeh et al. [60] find that supposedly-interpretable models can lead to a decreased ability to detect and correct model mistakes, possibly due to information overload. Jesus et al. [35] introduce a method to compare explanation methods based on how subject matter experts perform on specific tasks with the help of explanations. Lage et al. [44] use insights from rigorous human-subject experiments to inform regularizers used in explanation algorithms. In contrast to the aforementioned research, our work leverages twenty-two different state-of-the-art quantitative metrics to systematically benchmark the reliability (and not interpretability) of post hoc explanation methods.

**Limitations and Vulnerabilities of Post hoc Explanations.** Various quantitative metrics proposed in literature (See Section 1) were also leveraged to analyze the behavior of post hoc explanation methods and their vulnerabilities—e.g., Ghorbani et al. [29] and Slack et al. [68] demonstrated that methods such as LIME and SHAP may result in explanations that are not only inconsistent and unstable, but also prone to adversarial attacks. Furthermore Lakkaraju and Bastani [45] and Slack et al. [68] showed that explanations which do not accurately represent the importance of sensitive attributes (e.g., race, gender) could potentially mislead end users into believing that the underlying models are fair when they are not [45, 68, 6]. This, in turn, could lead to the deployment of unfair models in critical real world applications. There is also some discussion about whether models which are not inherently interpretable ought to be used in high-stakes decisions at all. Rudin [64] argues that post hoc explanations tend to be unfaithful to the model to the extent that their usefulness is severely compromised. While this line of work demonstrates different ways in which explanations could potentially induce inaccuracies and biases in real world applications, they do not focus on systematic benchmarking of post hoc explanation methods which is the main goal of our work.

## C  Synthetic Dataset

Our proposed data generation process described in Algorithm 1 is designed to encapsulate arbitrary feature dependencies, unambiguous definitions of local neighborhoods, and clear descriptions of feature influences. In our algorithm 1, the local neighborhood of a sample is controlled by its cluster membership $k$, while the degree of feature dependency can be easily controlled by the user setting $\Sigma_k$ to desired values. The default value of $\Sigma_k = \mathbf{I}$, where $\mathbf{I}$ is the identity matrix, indicating that all features are independent of one another. The elements of the true underlying weight vector $w_k$ are

sampled from a uniform distribution: $w_k \sim \mathcal{U}(l, u)$. We set $l = -1$ and $u = 1$. The masking vectors $m_k$ with elements $m_{k,j}$ are generated by a Bernoulli distribution: $m_{k,j} \sim \mathcal{B}(p)$, where the parameter $p$ controls the ground-truth explanation sparsity. We set $p = 0.25$. Further, the cluster centers are chosen as follows: when the number of features $d$ is smaller than or equal to the number of clusters $K$, then we set the first cluster center to $[1, 0, \ldots, 0]$, the second one to $[0, 1, \ldots, 0]$, etc. We also introduce a multiplier $\kappa$ to control the distance between the cluster centers so that the cluster centers are located at $\mu_1 = \kappa \cdot [1, 0, \ldots, 0]$, $\mu_2 = \kappa \cdot [0, 1, \ldots, 0]$, etc. We set $\kappa = 6$. In general, we have observed that lower values of $\kappa$ result in more complex classification problems. When the number of features $d$ is greater than the number of clusters $K$, we adjust the above described approach slightly: we first compute $\ell = \lfloor K/d \rfloor$, and then we compute the cluster centers for the first $d$ clusters as described above, for the next $d$ clusters we use $\mu_{d+1} = 2\kappa \cdot [1, 0, \ldots, 0]$, $\mu_{d+2} = 2\kappa \cdot [0, 1, \ldots, 0]$, etc. If $\ell = 1$, we stop this procedure here, else we continue, and repeat filling up the clusters $\mu_{2d+1} = 3\kappa \cdot [1, 0, \ldots, 0]$, $\mu_{2d+2} = 3\kappa \cdot [0, 1, \ldots, 0]$, etc.

---

**Algorithm 1: SYNTHGAUSS**

---

**input** : number of clusters: $K$, cluster centers: $[\mu_1, \mu_2, \ldots, \mu_K]$, cluster variances: $[\Sigma_1, \ldots, \Sigma_K]$, Weight vectors: $[w_1, \ldots, w_K]$, masking vectors: $[m_1, \ldots, m_K]$
**output :** features: $\mathbf{X}$, labels: $\mathbf{y}$
$\mathbf{X} = \mathbf{0}_{n \times d}$ ;
$\Pi = \mathbf{0}_{n \times 1}$ ;
**for** $i = 1{:}n$ **do**
    $k \leftarrow \mathrm{Cat}(K)$    # Randomly picks a cluster index ;
    $x_i \sim \mathcal{N}(\mu_k, \Sigma_k)$    # Samples Gaussian instance ;
    $\mathbf{X}[i, :] = x_i$ ;
    $\pi_1 = \mathbb{P}(y_i = 1 | \mathbf{x}_i) = \frac{\exp\left((w_k \odot m_k)^\top x_i\right)}{1 + \exp\left((w_k \odot m_k)^\top x_i\right)}$    # Get class probability ;
    $\Pi[i] = \pi_1$ ;
$\tilde{\pi} = \texttt{get\_median}(\Pi)$ ;
**for** $i = 1{:}n$ **do**
    $y_i = \mathbb{I}(\Pi[i] > \tilde{\pi})$    # Make sure classes are balanced ;
Return: $\mathbf{X}, \mathbf{y}$ ;

---

**Theorem 1** *If a given dataset encapsulates the properties of feature independence, unambiguous and well-separated local neighborhoods, and a unique ground truth explanation for each local neighborhood, the most accurate model trained on such a dataset will adhere to the unique ground truth explanation of each local neighborhood.*

*Proof:* Before we describe the proof, we begin by first formalizing the properties of feature independence, unambiguous and well-separated local neighborhoods, and a unique ground truth explanation for each local neighborhood using some notation. Let $A = [a_1, a_2, \cdots a_d]$ denote the vector of input features in a given dataset $D$, and let the instances in the dataset $D$ be separated into $K$ clusters (local neighborhoods) based on their proximity.

Feature independence: The dataset $D$ satisfies feature independence if $P(a_i|a_j) = P(a_i) \, \forall i, j \in \{1, 2, \cdots d\}$ and $i \neq j$.

Unambiguous and well-separated local neighborhoods: The dataset $D$ is said to constitute unambiguous and well-separated local neighborhoods if $dist(x_{i,j}, x_{p,q}) \gg dist(x_{i,j}, x_{i,l}) \, \forall i, p \in \{1, 2 \cdots K\}$ where $x_{i,j}$ and $x_{i,l}$ are the $j^{th}$ and $l^{th}$ instances of some cluster $i$, $x_{p,q}$ is the $q^{th}$ instance of cluster $p$ and $p \neq i$. The above implies that the distances between points in different clusters should be significantly higher than the distances between points in the same cluster.

Unique ground truth explanation for each local neighborhood: The dataset $D$ is said to comprise of unique ground truth explanations for each local neighborhood if the ground truth labels of instances in each cluster $k \in \{1, 2 \cdots K\}$ are generated as a function of a subset of features $A_k \subseteq A$ where there is a clear relative ordering among features in $A_k$ which is captured by the weight vector $w_k$, and features in $A$ are completely independent of each other. [Note that the influential feature set $A_k$ corresponding to the cluster $k$ is also captured using the mask vector $m_k$ (See Section 2) where

**Figure 4: A snapshot of the OpenXAI leaderboard submission form.** Users can use this single form to incorporate new datasets, pre-trained models, explanation methods, and evaluation metrics into OpenXAI framework.

$m_{k,i} = 1$ if the $i^{th}$ feature is in set $A_k$]. This would imply that there is a clear description of the top-$T$ features (and their relative ordering) where $T \le |A_k|$ influence the ground truth labeling process.

With the above information in place, let us now consider a model $\mathcal{M}$ which is trained on the dataset $D$ and achieves highest possible accuracy on it. To show that this model indeed adheres to the unique ground truth explanations of each of the local neighborhoods (clusters), we adopt the strategy of proof by contradiction. To this end, we begin with the assumption that the model $\mathcal{M}$ does not adhere to the unique ground truth explanation of some local neighborhood $k \in \{1, 2, \cdots K\}$ i.e., the top-$T$ features leveraged by model $\mathcal{M}$ for instances in cluster $k$ (denoted by $\mathcal{M}_T^k$) do not exactly match the top-$T$ features leveraged by the ground truth labeling process where $T \le |A_k|$. This happens either when there is a mismatch in the relative ordering among the top-$T$ features used by the model $\mathcal{M}$ and the ground truth labeling process (or) when there is at least one feature $a$ that appears among the top-$T$ features used by the model but not the ground truth labeling process. In either case, we can construct another model $\mathcal{M}'$ such that the top-$T$ features used by $\mathcal{M}'$ match the top-$T$ features of the ground truth labeling process exactly, and the accuracy of $\mathcal{M}'$ will be higher than that of $\mathcal{M}$. This contradicts the assumption that the model $\mathcal{M}$ has the highest possible accuracy, thus demonstrating that the model with highest possible accuracy would adhere to the unique ground truth explanation of each local neighborhood.

# D  Extending OpenXAI

In this section, we walk through how users can extend and contribute to OpenXAI by adding custom datasets, pre-trained models, explanation methods, or evaluation metrics. In Figure 4, we show a snapshot of our FORM that can be used by users to submit requests for new datasets, pre-trained models, explanation methods, or evaluation metrics.

**Custom Datasets.** Adding new datasets into OpenXAI 's pipeline is as simple as uploading a .csv file or a .zip folder. Users can submit requests to incorporate new datasets into OpenXAI by

providing either the complete dataset (with or without data splits) or individual training and testing files for the respective dataset.

**Custom Predictive Models.** Users can also submit requests to include new pre-trained models into the OpenXAI framework. Here, users have to provide i) model architecture and implementation details, ii) model hyperparameters and training details, and iii) trained weights for reproducing model performance.

**Custom Evaluation Metrics.** Users can include new evaluation metrics into OpenXAI by filling out the submit request form and providing the GitHub link to their code and a summary of their metrics. Once approved, the metric should be implemented as a function in the `Evaluator` class. The input data, ground-truth labels, model predictions, explanations, and the underlying black-box model are all provided in the `Evaluator` class. The evaluation metric should return a scalar score as a part of their respective implementation.

**Custom Explanation Methods.** Users can include new explanation methods into OpenXAI 's list of explainers by following the OPENXAI.EXPLAINER template. The abstract method defined ensures that the output of new/existing explanation methods returns explanations as a tensor. Note that all custom explanation methods should extend the `Explainer` class. Finally, users must provide the GitHub link to their code and a summary of their explanation method.

We invite submissions to any one or multiple benchmarks in OpenXAI. To be included in the leaderboard, please fill out this FORM, include results of your explanation method and provide the full implementation of your algorithm with proper documentations and GitHub link.

# E   Benchmarking Analysis

## E.1   Real-World Datasets

In addition to synthetic dataset, OpenXAI library includes five real-world benchmark datasets from high-stakes domains. Table 1 provides a summary of the real-world datasets currently included in OpenXAI. Our library implements multiple data split strategies and allows users to customize the percentages of train-test splits. To this end, if a given dataset comes with a pre-determined train and test splits, OpenXAI loads the training and testing dataloaders from those pre-determined splits. Otherwise, OpenXAI's dataloader divides the entire dataset randomly into train (70%) and test (30%). Next, we detail the covariates $\mathbf{x}$ and labels $\mathbf{y}$ for each dataset.

**German Credit.** The dataset comprises of demographic (age, gender), personal (marital status), and financial (income, credit duration) features from 1,000 credit applicants, where they are categorized into good vs. bad customer depending on their credit risk [22].

**COMPAS.** The dataset has criminal records and demographics features for 18,876 defendants who got released on bail at the U.S state courts during 1990-2009. The task is to classify defendants into bail (i.e., unlikely to commit a violent crime if released) vs. no bail (i.e., likely to commit a violent crime) [36].

**Adult Income.** The dataset contains demographic (e.g., age, race, and gender), education (degree), employment (occupation, hours-per week), personal (marital status, relationship), and financial (capital gain/loss) features for 48,842 individuals. The task is to predict whether an individual's income exceeds $50K per year vs. not [79].

**Give Me Some Credit.** The dataset incorporates demographic (age), personal (number of dependents), and financial (e.g., monthly income, debt ratio, etc.) features for 250,000 individuals. The task is to predict the probability that a customer will experience financial distress vs. not in the next two years. The aim of the dataset is to build models that customers can use to the best financial decisions [27].

**Home Equity Line of Credit (HELOC).** The dataset comprises of financial (e.g., total number of trades, average credit months in file) attributes from anonymized applications submitted by 9,871 real homeowners. A HELOC is a line of credit typically offered by a bank as a percentage of home equity. The fundamental task is to use the information about the applicant in their credit report to predict whether they will repay their HELOC account within 2 years [25].

## E.2 Explanation Methods

OpenXAI provides implementations for six explanation methods: Vanilla Gradients [67], Integrated Gradients [72], SmoothGrad [70], Gradient x Input [66], LIME [62], and SHAP [55]. Table 6 summarizes how these methods differ along two axes: whether each method requires "White-box Access" to model gradients and whether each method learns a local approximation model to generate explanations.

**Table 6:** Summary of currently available feature attribution methods in OpenXAI. "Learning?" denotes whether learning/training procedures are required to generate explanations and "White-box Access?" denotes if the access to the internals of the model (e.g., gradients) are needed.

| Method | Learning? | White-box Access? |
|---|---|---|
| Random | ✗ | ✗ |
| VanillaGrad | ✗ | ✓ |
| IntegratedGrad | ✗ | ✓ |
| Gradient x Input | ✗ | ✓ |
| SmoothGrad | ✗ | ✓ |
| SHAP | ✓ | ✗ |
| LIME | ✓ | ✗ |

**Implementations.** We used existing public implementations of all explanation methods in our experiments. We used the following `captum` software package classes: i) `captum.attr.Saliency` for Vanilla Gradients; ii) `captum.attr.IntegratedGradients` for Integrated Gradients; iii) `captum.attr.NoiseTunnel` and `captum.attr.Saliency` for SmoothGrad; iv) `captum.attr.InputXGradient` for Gradient x Input; and v) `captum.attr.KernelShap` for SHAP. Finally, we use the authors' LIME python package for LIME.

## E.3 Hyperparameter details

OpenXAI uses default hyperparameter settings for all explanation methods following the authors' guidelines. Every explanation method has a corresponding parameter dictionary that stores all the default parameter values. Below, we detail the hyperparameters of individual explanation methods used in our experiments.

**a) params_lime** = {'lime_mode': 'tabular', 'lime_sample_around_instance': True, 'lime_kernel_ width': 0.75, 'lime_n_samples': 1000, 'lime_discretize_continuous': False, 'lime_standard_ deviation': float(np.sqrt(0.05))}

*Description.* The option 'tabular' indicates that we wish to compute explanations on a tabular data set. The option 'lime_sample_around_instance' makes sure that the sampling is conducted in a local neighborhood around the point that we wish to explain. The 'lime_discretize_continuous' option ensures that continuous variables are kept continuous, and are not discretized. This parameter sets the 'lime_standard_ deviation' of the Gaussian random variable that is used to sample around the instance that we wish to explain. We set this to a small value, ensuring that we in fact sample from a local neighborhood.

**b) params_shap** = {'shap_subset_size': 50, 'perturbations_per_eval': 1, 'feature_mask': None, 'baselines': None}

*Description.* The parameter 'shap_subset_size' controls the number of samples of the original model used to train the surrogate SHAP model, 'perturbations_per_eval' allows the processing of multiple samples simultaneously, 'feature_mask' defines a mask on the input instance that group features corresponding to the same interpretable feature, and 'baselines' defines the reference value which replaces each feature when the corresponding interpretable feature is set to 0.

**c) params_grads** = {'grad_absolute_value': False}

*Description.* The parameter 'grad_absolute_value' controls whether the absolute value of each element in the explanation vector should be taken or not.

**d) params_sg** = {'sg_n_samples': 500, 'sg_standard_deviation': float(np.sqrt(0.05))}

*Description.* The parameter 'sg_n_samples' sets the number of samples used in the Gaussian random variable to smooth the gradient, while 'sg_standard_deviation' determines the size of the local neighboorhood the gaussian random variables are sampled from. We use a small value, ensuring that we in fact sample from a local neighborhood.

**e) params_ig** = {'ig_method': 'gausslegendre', 'ig_multiply_by_inputs': False, 'ig_baseline': 'mean'}

Further, we parameterized our data generating process as follows.

**f) params_gauss** = {'n_samples': 1000, 'dim':20, 'n_clusters': 10, 'distance_to_center': 6, 'test_size': 0.25, 'upper_weight': 1, 'lower_weight': -1, 'seed': 564, 'sigma': None, 'sparsity': 0.25}

*Description.* The above parameters can be matched with the parameters from the data generating process described in Appendix C: in particular, 'sparsity'= $p$, 'upper_weight' = $u$, 'lower_weight' = $l$, 'dim'= $d$, 'n_clusters' = $K$, 'distance_to_center' = $\kappa$ and 'sigma'=None implies that the identity matrix is used, i.e., $\Sigma_k = \mathbf{I}$.

In addition, for a given instance $\mathbf{x}$, some evaluation metrics leverage a perturbation class to generate perturbed samples $\mathbf{x}'$. We have a parameterized version of this perturbation class in OpenXAI with the following default parameters:

**g) params_perturb** = {'perturbation_mean' : 0.0, 'perturbation_std' : 0.05, 'perturbation_flip_percentage' : 0.03}

Finally, we denote the top-$k$ value using {'percentage_most_important': 0.25}, i.e., we consistently use 25% of the top features for calculating our metric scores.

## E.4 Model details

Our current release of OpenXAI has two pre-trained models: i) a logistic regression model and ii) a deep neural network model for all datasets. To support systematic, reproducible, and efficient benchmarking of post hoc explanation methods, we provide the model weights for both models trained on all six datasets in our pipeline. For neural network models, we use two fully connected hidden layers with 100 nodes in each layer, with ReLU activation functions and an output softmax layer for all datasets. We train both models for 50 epochs using an Adam optimizer with a learning rate of $\eta$=0.001. Next, we show the model performance of both models on all six datasets.

**Table 7: Results of the machine learning models trained on six datasets.** Shown are the accuracy of LR and ANN models trained the datasets. The best performance is bolded.

| Dataset | LR | ANN |
|---|---|---|
| Synthetic Data | 83.0% | **92.0%** |
| German Credit | 72.0% | **75.0%** |
| HELOC | 72.0% | **74.0%** |
| COMPAS | **85.4%** | 85.0% |
| Adult Income | 84.0% | **85.0%** |
| Give Me Some Credit | 94.1% | **95.0%** |
| Framingham Heart Study | 85.0% | 85.0% |
| Pima-Indians Diabetes | 66.0% | **77.0%** |

## E.5 Additional results

The datasets included in our OpenXAI framework are diverse i.e., these datasets span a wide variety of sizes (ranging from 1000 instances to 102,209 instances in the dataset), feature dimensions (ranging from 7 to 23 dimensions), class imbalance ratios, and feature types (comprising a mix of continuous and discrete features). In addition, the datasets that we chose span two different real-world domains namely criminal justice, and lending. In addition, the datasets included in our OpenXAI framework are very popular and are widely employed in XAI and fairness research. For instance, several recent works in XAI have employed these datasets to both evaluate the efficacy of newly proposed methods as well as to study the behavior of existing methods. Furthermore, several of the datasets that we chose also include sensitive attributes which help us evaluate the (un)fairness (disparities) in explanation quality across majority and minority subgroups.

In response to the reviewer's comment about the lack of diversity in the domains we consider, we included two new datasets from the healthcare domain and benchmarked state-of-the-art explanation methods on these datasets as well. More specifically, we included the Pima-Indians Diabetes [71] and Framingham heart study [1] datasets both of which have been utilized in recent XAI research. New results with these datasets are included in Section E.5 in the Appendix. We observe similar insights with these new datasets as well. We find that explanations constructed by SmoothGrad are on average more faithful (see Tables 8-11 in Appendix) and outperform other feature-attribution methods on the Prediction Gap on Unimportant feature perturbation (PGU) metric. The stability results of various explanation methods for logistic regression and neural network models are shown in Tables 8-9 and Tables 10-11 respectively in the Appendix. For logistic regression models, we do not find consistent trends across both the new datasets. On the other hand, in the case of neural network models, we observe that SmoothGrad, on average, outperforms other feature attribution methods on relative representation stability (RRS) and output stability (ROS) metrics.

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

| Method | RIS | RRS | ROS |
|---|---|---|---|
| Random | 4.56±0.01 | 8.08±0.04 | 8.08±0.04 |
| VanillaGrad | **0.03**±0.02 | 1.72±0.04 | 1.72±0.04 |
| IntegratedGrad | -0.56±0.01 | **1.14**±0.04 | **1.14**±0.04 |
| Gradient x Input | -0.37±0.02 | 2.53±0.05 | 2.53±0.05 |
| SmoothGrad | -4.52±0.02 | -2.35±0.04 | -2.35±0.04 |
| SHAP | -0.76±0.02 | 2.42±0.05 | 2.42±0.05 |
| LIME | -0.84±0.01 | 1.69±0.04 | 1.69±0.04 |

**Table 13: Stability of explanation methods on the Pima Indians Diabetes dataset with LR model.** Shown are average and standard error values across all test set instances. Values closer to zero are desirable, and the best performance is bolded.

| Method | RIS | RRS | ROS |
|---|---|---|---|
| Random | 7.68±0.14 | 7.32±0.10 | 7.32±0.10 |
| VanillaGrad | 2.30±0.13 | 0.35±0.15 | 0.35±0.15 |
| IntegratedGrad | 1.52±0.12 | -0.43±0.15 | -0.43±0.15 |
| Gradient x Input | 2.61±0.13 | 2.43±0.10 | 2.43±0.10 |
| SmoothGrad | -2.78±0.14 | -3.98±0.12 | -3.98±0.12 |
| SHAP | 2.45±0.13 | 2.31±0.10 | 2.31±0.10 |
| LIME | **1.46**±0.14 | **-0.04**±0.12 | **-0.04**±0.12 |

**Table 14: Stability of explanation methods on the Framingham heart study dataset with ANN model.** Shown are average and standard error values across all test set instances. Values closer to zero are desirable, and the best performance is bolded.

| Method | RIS | RRS | ROS |
|---|---|---|---|
| Random | 4.58±0.01 | 7.35±0.02 | 13.34±0.05 |
| VanillaGrad | 2.95±0.03 | 5.18±0.04 | 10.80±0.07 |
| IntegratedGrad | 1.94±0.03 | 4.32±0.03 | 10.12±0.05 |
| Gradient x Input | 1.89±0.04 | 4.09±0.04 | 9.77±0.07 |
| SmoothGrad | -4.15±0.02 | **-1.49**±0.02 | **4.43**±0.05 |
| SHAP | **0.74**±0.02 | 3.38±0.03 | 9.31±0.05 |
| LIME | 1.03±0.01 | 3.64±0.02 | 9.63±0.05 |

**Table 15: Stability of explanation methods on the Pima Indians Diabetes dataset with ANN model.** Shown are average and standard error values across all test set instances. Values closer to zero are desirable, and the best performance is bolded.

| Method | RIS | RRS | ROS |
|---|---|---|---|
| Random | 7.60±0.14 | 7.00±0.05 | 12.75±0.12 |
| VanillaGrad | 5.24±0.18 | 4.24±0.12 | 9.76±0.17 |
| IntegratedGrad | 5.19±0.15 | 4.25±0.06 | 9.92±0.12 |
| Gradient x Input | 4.33±0.19 | 3.31±0.12 | 8.88±0.17 |
| SmoothGrad | **-1.18**±0.14 | **-1.90**±0.05 | **3.85**±0.11 |
| SHAP | 3.32±0.18 | 2.58±0.10 | 8.17±0.15 |
| LIME | 3.82±0.14 | 3.09±0.06 | 8.90±0.12 |

## E.6 Additional Results on LR models

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

| Method | RIS | RRS |
|---|---|---|
| Random | 5.387±0.020 | 13.794±0.029 |
| VanillaGrad | 2.124±0.019 | 6.336±0.015 |
| IntegratedGrad | 1.403±0.019 | 5.776±0.008 |
| Gradient x Input | **1.036**±0.022 | 8.964±0.036 |
| SmoothGrad | -3.724±0.023 | **3.507**±0.029 |
| SHAP | 1.458±0.021 | 9.331±0.030 |
| LIME | 4.534±0.019 | 8.833±0.029 |

**Table 20: Stability of explanation methods on the Adult Income dataset with LR model.** Shown are average and standard error values across all test set instances. Values closer to zero are desirable, and the best performance is bolded.

| Method | RIS | RRS |
|---|---|---|
| Random | 6.345±0.009 | 12.577±0.018 |
| VanillaGrad | 4.092±0.018 | 6.928±0.012 |
| IntegratedGrad | 3.492±0.013 | 6.843±0.014 |
| Gradient x Input | 2.307±0.016 | 7.543±0.019 |
| SmoothGrad | -3.007±0.009 | **2.941**±0.019 |
| SHAP | 1.857±0.010 | 7.883±0.019 |
| LIME | **1.715**±0.009 | 7.899±0.016 |

**Table 21: Stability of explanation methods on the COMPAS dataset with LR model.** Shown are average and standard error values across all test set instances. Values closer to zero are desirable, and the best performance is bolded.

| Method | RIS | RRS | ROS |
|---|---|---|---|
| Random | 7.88±0.04 | 6.74±0.05 | 6.74±0.05 |
| VanillaGrad | 4.77±0.05 | 1.82±0.05 | 1.82±0.05 |
| IntegratedGrad | 4.33±0.04 | 1.41±0.04 | 1.41±0.04 |
| Gradient x Input | 2.98±0.06 | 1.78±0.06 | 1.78±0.06 |
| SmoothGrad | -1.10±0.03 | **-2.93**±0.04 | **-2.93**±0.04 |
| SHAP | 3.12±0.05 | 2.35±0.05 | 2.35±0.05 |
| LIME | **4.24**±0.04 | 2.28±0.04 | 2.28±0.04 |

**Table 22: Stability of explanation methods on the GMSC Credit dataset with LR model.** Shown are average and standard error values across all test set instances. Values closer to zero are desirable, and the best performance is bolded.

| Method | RIS | RRS | ROS |
|---|---|---|---|
| Random | 5.52±0.007 | 5.97±0.009 | 5.97±0.009 |
| VanillaGrad | 3.96±0.008 | 1.38±0.007 | 1.38±0.007 |
| IntegratedGrad | 3.12±0.008 | 0.59±0.007 | 0.59±0.007 |
| Gradient x Input | 2.06±0.009 | 0.15±0.009 | 0.15±0.009 |
| SmoothGrad | -3.20±0.005 | **-3.56**±0.008 | **-3.56**±0.008 |
| SHAP | 1.44±0.008 | 0.09±0.009 | 0.09±0.009 |
| LIME | **2.56**±0.006 | 1.81±0.008 | 1.81±0.008 |

## E.7   Results on ANN models

**Table 23: Predicted faithfulness results on the SynthGauss dataset for all explanation methods with ANN model.** Shown are average and standard error metric values computed across all instances in the test set. ↑ indicates that higher values are better, and ↓ indicates that lower values are better. Values corresponding to best performance are bolded.

| Method | PGU (↓) | PGI (↑) |
|---|---|---|
| Random | 0.054±0.002 | 0.06±0.002 |
| VanillaGrad | 0.059±0.002 | 0.061±0.002 |
| IntegratedGrad | 0.059±0.002 | 0.061±0.002 |
| Gradient x Input | 0.059±0.002 | 0.061±0.002 |
| SmoothGrad | **0.028**±0.001 | **0.08**±0.003 |
| SHAP | 0.057±0.002 | 0.062±0.002 |
| LIME | 0.069±0.003 | 0.051±0.002 |

**Table 24: Predicted faithfulness results on the German dataset for all explanation methods with ANN model.** Shown are average and standard error metric values computed across all instances in the test set. ↑ indicates that higher values are better, and ↓ indicates that lower values are better. Values corresponding to best performance are bolded.

| Method | PGU (↓) | PGI (↑) |
|---|---|---|
| Random | 0.067±0.004 | 0.02±0.002 |
| VanillaGrad | 0.064±0.004 | 0.022±0.002 |
| IntegratedGrad | 0.064±0.004 | 0.022±0.002 |
| Gradient x Input | **0.059**±0.005 | 0.03±0.002 |
| SmoothGrad | 0.062±0.004 | 0.023±0.002 |
| SHAP | 0.06±0.004 | **0.027**±0.002 |
| LIME | 0.063±0.004 | 0.023±0.002 |

**Table 25: Predicted faithfulness results on the Heloc dataset for all explanation methods with ANN model.** Shown are average and standard error metric values computed across all instances in the test set. ↑ indicates that higher values are better, and ↓ indicates that lower values are better. Values corresponding to best performance are bolded.

| Method | PGU (↓) | PGI (↑) |
|---|---|---|
| Random | 0.056±0.001 | 0.059±0.001 |
| VanillaGrad | 0.061±0.001 | 0.059±0.001 |
| IntegratedGrad | 0.061±0.001 | 0.059±0.001 |
| Gradient x Input | 0.056±0.001 | 0.06±0.001 |
| SmoothGrad | **0.032**±0.001 | **0.078**±0.001 |
| SHAP | 0.055±0.001 | 0.061±0.001 |
| LIME | 0.057±0.001 | 0.062±0.001 |

**Table 26: Predicted faithfulness results on the Adult Income dataset for all explanation methods with ANN model.** Shown are average and standard error metric values computed across all instances in the test set. ↑ indicates that higher values are better, and ↓ indicates that lower values are better. Values corresponding to best performance are bolded.

| Method | PGU (↓) | PGI (↑) |
|---|---|---|
| Random | 0.055±0.001 | 0.063±0.001 |
| VanillaGrad | 0.064±0.001 | 0.053±0.001 |
| IntegratedGrad | 0.066±0.001 | 0.051±0.001 |
| Gradient x Input | 0.055±0.001 | 0.062±0.001 |
| SmoothGrad | **0.014**±0.001 | **0.100**±0.001 |
| SHAP | 0.049±0.001 | 0.070±0.001 |
| LIME | 0.019±0.001 | 0.098±0.001 |

**Table 27: Stability of explanation methods on the SynthGauss dataset with ANN model.** Shown are average and standard error values across all test set instances. Values closer to zero are desirable, and the best performance is bolded.

| Method | RIS | RRS | ROS |
|---|---|---|---|
| Random | 9.35±0.01 | 7.37±0.02 | 13.81±0.10 |
| VanillaGrad | 7.58±0.09 | 4.24±0.10 | 8.02±0.07 |
| IntegratedGrad | 8.14±0.03 | 5.53±0.03 | 11.60±0.08 |
| Gradient x Input | 6.80±0.09 | 3.46±0.10 | 7.22±0.07 |
| SmoothGrad | **1.11**±0.01 | **-1.07**±0.02 | **5.31**±0.10 |
| SHAP | 5.96±0.01 | 3.95±0.02 | 10.39±0.09 |
| LIME | 5.53±0.01 | 3.40±0.02 | 9.86±0.09 |

**Table 28: Stability of explanation methods on the German Credit dataset with ANN model.** Shown are average and standard error values across all test set instances. Values closer to zero are desirable, and the best performance is bolded.

| Method | RIS | RRS | ROS |
|---|---|---|---|
| Random | 1.61±0.24 | 3.89±0.28 | 7.65±0.17 |
| VanillaGrad | 1.35±0.10 | 4.44±0.10 | 11.35±0.36 |
| IntegratedGrad | **0.62**±0.23 | 2.88±0.28 | **6.83**±0.17 |
| Gradient x Input | -1.90±0.08 | **1.32**±0.06 | 8.10±0.36 |
| SmoothGrad | 2.97±0.07 | 6.17±0.05 | 13.10±0.33 |
| SHAP | 1.28±0.08 | 4.46±0.07 | 11.46±0.30 |
| LIME | 6.35±0.07 | 9.57±0.04 | 16.50±0.34 |

**Table 29: Stability of explanation methods on the Heloc dataset with ANN model.** Shown are average and standard error values across all test set instances. Values closer to zero are desirable, and the best performance is bolded.

| Method | RIS | RRS | ROS |
|---|---|---|---|
| Random | 5.39±0.02 | 7.63±0.03 | 10.99±0.04 |
| VanillaGrad | 4.06±0.04 | 5.86±0.03 | 10.80±0.04 |
| IntegratedGrad | 3.16±0.02 | 5.17±0.01 | 9.53±0.04 |
| Gradient x Input | 2.46±0.04 | 4.28±0.03 | 9.56±0.04 |
| SmoothGrad | -3.11±0.02 | **-0.95**±0.01 | **4.60**±0.04 |
| SHAP | 1.57±0.02 | 3.80±0.01 | 9.40±0.04 |
| LIME | **1.19**±0.02 | 3.33±0.01 | 13.25±0.04 |

**Table 30: Stability of explanation methods on the Adult Income dataset with ANN model.** Shown are average and standard error values across all test set instances. Values closer to zero are desirable, and the best performance is bolded.

| Method | RIS | RRS | ROS |
|---|---|---|---|
| Random | 6.34±0.01 | 7.27±0.009 | 14.38±0.05 |
| VanillaGrad | 5.01±0.04 | 4.58±0.042 | 9.857±0.02 |
| IntegratedGrad | 4.32±0.01 | 4.33±0.011 | 11.74±0.05 |
| Gradient x Input | 3.25±0.04 | 3.08±0.041 | 8.248±0.02 |
| SmoothGrad | -1.70±0.01 | **-0.95**±0.010 | **6.222**±0.06 |
| SHAP | 1.98±0.01 | 2.68±0.013 | 9.780±0.05 |
| LIME | **1.79**±0.01 | 2.45±0.008 | 9.773±0.05 |

**Table 31: Stability of explanation methods on the COMPAS dataset with ANN model.** Shown are average and standard error values across all test set instances. Values closer to zero are desirable, and the best performance is bolded.

| Method | RIS | RRS | ROS |
|---|---|---|---|
| Random | 7.78±0.20 | 7.21±0.17 | 11.51±0.18 |
| VanillaGrad | 6.29±0.24 | 5.54±0.14 | 9.02±0.31 |
| IntegratedGrad | 5.91±0.17 | 5.29±0.10 | 9.27±0.19 |
| Gradient x Input | 3.81±0.29 | 3.19±0.16 | 7.16±0.32 |
| SmoothGrad | **-0.59**±0.17 | **-1.39**±0.13 | **3.15**±0.14 |
| SHAP | 4.24±0.18 | 3.72±0.10 | 8.27±0.17 |
| LIME | 4.49±0.19 | 3.54±0.09 | 8.36±0.17 |

**Table 32: Stability of explanation methods on the GMSC Credit dataset with ANN model.** Shown are average and standard error values across all test set instances. Values closer to zero are desirable, and the best performance is bolded.

| Method | RIS | RRS | ROS |
|---|---|---|---|
| Random | 5.52±0.006 | 7.78±0.004 | 12.07±0.009 |
| VanillaGrad | 3.91±0.008 | 5.63±0.006 | 8.46±0.014 |
| IntegratedGrad | 3.07±0.007 | 4.76±0.004 | 7.82±0.0115 |
| Gradient x Input | 1.89±0.008 | 3.54±0.007 | 6.67±0.0139 |
| SmoothGrad | -3.03±0.006 | **-1.02**±0.004 | **3.20**±0.009 |
| SHAP | 1.27±0.007 | 3.06±0.004 | 6.93±0.010 |
| LIME | **1.25**±0.006 | 3.31±0.004 | 7.82±0.009 |

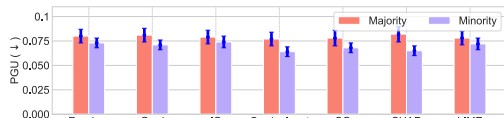

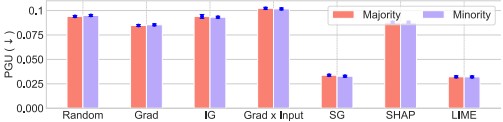

**Figure 5: Fairness analysis of PGU metric on the German Credit dataset with ANN model.** Shown are average and standard error values for majority (male) and minority (female) subgroups. Larger gaps between the values of majority and minority subgroups (i.e, red and blue bars respectively) indicate higher disparities which are undesirable.

**Figure 6: Fairness analysis of PGU metric on the Adult Income dataset with ANN model.** Shown are average and standard error values for majority (male) and minority (female) subgroups. Larger gaps between the values of majority and minority subgroups (i.e, red and blue bars respectively) indicate higher disparities which are undesirable.

# F    Choice of XAI methods, datasets, and models

While feature attribution-based explanation methods such as LIME, SHAP, and Gradient-based methods have been proposed a few years back, they continue to be the most popular and widely used post hoc explanation methods both in research [16, 43, 7, 33, 31] and in practice [23, 78, 34, 28]. In fact, several recent works published in 2022 have analyzed these methods both theoretically and empirically, and have called for further study of these methods given their widespread adoption [19, 16, 9, 31, 43, 26]. Furthermore, recent research has also argued that there is little to no understanding of the behavior and effectiveness of even basic post hoc explanation methods such as LIME, SHAP, and gradient-based methods [43, 50, 64, 40, 10, 3], and developing such an understanding would be a critical first step towards the progress of the XAI field. To this end, we focus on these methods for the first release of our OpenXAI framework. In the next release, we plan to evaluate and benchmark other recently proposed methods (e.g., TCAV and its extensions, influence functions, etc.) as well.

Note that the evaluation metrics and the explanation methods that we include in our framework are generic enough to be applicable to other modalities of data including text and images. The reason why we focused on tabular data for the first release of OpenXAI is two-fold: i) The need for model understanding and explainability is often motivated by high-stakes decision-making settings and applications e.g., loan approvals, disease diagnosis and treatment recommendations, recidivism prediction etc. [14, 57, 58]. Data encountered in these settings is predominantly tabular. ii) Recent research has argued that there is no clear understanding as to which explanation methods perform well on what kinds of metrics even on simple, low-dimensional tabular datasets [50, 19, 43, 31], and that this is a big open question which has far reaching implications for the progress of the field. To this end, we focused on tabular data for the first release of OpenXAI so that we could find some answers to the aforementioned question in the context of simple, low-dimensional tabular datasets before proceeding to high-dimensional image and text datasets. In the next release of OpenXAI, we plan to include and support image and text datasets, and also add metrics and explanation methods that are specific to these new data modalities.

The datasets that we utilize in this work are very popular and are widely employed in XAI and fairness research till date. For instance, several recent works in XAI published at ICML, NeurIPS, and FAccT conferences in 2021-22 have employed these datasets both to evaluate the efficacy of newly proposed methods, as well as to study the behavior of existing methods [9, 18–20, 38, 69, 73, 5]. Given this, we follow suit and employ these datasets in our benchmarking efforts. Similarly, the aforementioned works also employ logistic regression models and deep neural network architectures similar to the ones considered in our research.