# OpenReview forum: "OpenXAI: Towards a Transparent Evaluation of Model Explanations"
_NeurIPS.cc/2022/Track/Datasets_and_Benchmarks — NeurIPS 2022 Datasets and Benchmarks _

### Official Review · Reviewer_BXQ8 · 2022-07-13
**Benchmarking xai algorithm is still an open problem. But the proposed solution should be improved.**

**Rating:** 7
**Confidence:** 5

**Strengths:**

This benchmark is one of the first to propose a unification between various related works in order to standardize the evaluation of XAI. They did collect a good number of metrics compared to prior work.

The authors propose a new synthetic dataset to evaluate the xai algorithms in line 140. It is more elaborated than those reported in related work. It is just a bit confusing for the reader that the authors affirm in lines 103 and 104, that using synthetic data is not efficient in general. It would be more consistent to rephrase it.


**Weaknesses:**

openxai is not the first benchmark for XAI tools. The difference with prior work is not clearly defined. Comparing openxai to xai-bench, openxai is an improvement on xai-bench, but it is not a cutting-edge solution. Therefore, why not work on the same GitHub repository?

The second important point is the usability of the benchmark. It is quite challenging to go through this amount of scores and to compare even 2 XAI. How could an end-user exploit the benchmark and decide which xai tool to use?

Line 186: code snippets and tutorials on how to use the benchmark should be written in the GitHub repo or website but not in a research paper. The paper should focus on the advantages of the solution, i.e., proof of the efficiency of the benchmark in analyzing available xai.
Same for line 192 "Users can also submit requests to incorporate custom pre-trained models into the OpenXAI framework by filling a simple form and providing details about model architecture and parameters" and line 215...


According to the online interface (https://open-xai.github.io/leaderboard) end-users can submit a list of scores obtained using another repository. Therefore, authors cannot guarantee to provide a transparent benchmark, given that they have low control over the experiment code.
For example, end-user 1 creates a dataset and submits the obtained scores to open-xai then he updates the dataset, or it gets deleted.
Mainly, relying on other people's repo is only possible if they comply with the key criterion set by NeurIPS, which is accessibility: datasets should be available and accessible (see https://neurips.cc/Conferences/2022/CallForDatasetsBenchmarks).

Line 457: A more appropriate limitation of a benchmark would be the coverage of quantitative metrics only. It is indeed difficult to measure specific properties of the xai algorithms (for example, how the user interacts with the explanation). Another point to highlight in the web interface is the usage of default hyper-parameters (of the xai algorithms) and their influence on the leaderboard.
End-users that go through the provided website only should be informed of these limitations of the benchmark (or at least get redirected to the Limitation section in the paper) to avoid the bandwagon effect and to reduce potential over-trust of the benchmarked xai tools.


**Additional Feedback:**

the name "open-xai" does not really reflect the intended use of the framework. Authors should consider renaming it.

**Clarity:**

Authors should state in the paper's title that the method is made for post hoc xai or, more precisely, feature attribution xai, i.e., the title could be "OpenXAI: Towards a Transparent Evaluation of Post-hoc Model Explanations."

It would be helpful to mention in the abstract that this first version of the benchmark is restricted to tabular data

Figures 3 and 4: Please add an arrow on the y-axis to express "higher is better" or "lower is better."

Please include the main takeaway of the result discussion in the conclusion section.

The usage of paragraphs with titles made the article quite clear and readable.

Overall, the paper is understandable. It might even be redundant and thus excessively long. Please bare that the 9 pages limit is a maximum threshold and is not a selection criterion nor an excuse to reject papers.


**Correctness:**

Line 26: change "referred to as feature attribution methods." to "referred to as local feature attribution methods."

Line 10: (iii) the first ever public XAI leaderboards to benchmark explanations.
Many benchmarks for xai are already available online.
See, for example, SHAP benchmark https://shap.readthedocs.io/en/latest/example_notebooks/benchmarks/tabular/Benchmark%20XGBoost%20explanations.html
Or the xai bench [22]



**Documentation:**

line 13: "Overall, OpenXAI provides an automated end-to-end pipeline ..." In case there is one xai that provides correct results but does not fit into the proposed metric, dataset, or ML model, how would this be documented?


**Ethics:**

There are no ethical issues.


**Relation To Prior Work:**

Line 28: paper [30] is not a survey paper and does not actually analyze the trend

Line 93: Citation [3]
Add exactly which xai are failing the test or rephrase the statement to "some xai"


**Summary And Contributions:**

Openxai benchmarks 7 post-hoc feature attribution methods using 22 metrics.
Openxai collects existing metrics classified into faithfulness, stability, and fairness.
The project is open source, and anyone can add a metric or an XAI tool.

---

### Official Review · Reviewer_nA8i · 2022-07-26
**Good underlying concept, some room for improvement in implementation/analysis**

**Rating:** 7
**Confidence:** 5
**Clarity:** Yes

**Strengths:**

- The benchmark tool meets a genuine need not just in research, but for practicing data scientists. Prior research has shown that explanation methods do disagree substantially with each other, and data scientists do not have principled or consistent ways to adjudicate these disagreements.
- The tool incorporates diverse metrics in one package which is convenient for researchers who need to evaluate new explanation methods. Before researchers had to write their own code to evaluate their explanation methods, and had to read the literature to figure out which metrics to translate into code.
- The work has positive ethical and social implications because it aids the responsible use of explanation method outputs.
- The API of the evaluator tool makes a lot of sense and looks simple to use.

**Weaknesses:**

- The authors claim that the dataset choices are diverse, but 4 of the 5 datasets are about financial info, and all datasets contain demographic info.
- There's good arguments not to use Adult Income dataset because it lacks external validity (see https://proceedings.neurips.cc/paper/2021/hash/32e54441e6382a7fbacbbbaf3c450059-Abstract.html ). Similar arguments apply to the COMPAS dataset (https://arxiv.org/abs/2106.05498 ).
- Given the focus on tabular datasets, it's peculiar that the models provided are limited to logistic regression and neural networks. Tree ensemble methods often perform better than neural networks on tabular data. The accuracy figures on e.g. German Credit dataset are not particularly impressive. In addition, outside of the synthetic dataset, the neural networks are barely more accurate than the logistic regression, which sheds doubt on whether the neural networks are even picking up on correlations/interactions that the logistic regression is unable to. If they are not, then why do we need post-hoc explanation methods? The benchmark would be more interesting if it compared complex models like NNs and tree ensembles with LR in cases where the former have a real accuracy benefit, because then we can see how well the post-hoc explanations do when correlations/interactions really matter to the predictions.
- It's not ideal that the parameter choices for the SHAP method are so limited---the only parameter is the size of background data sample (it seems). Typical open-source SHAP packages provide many more parameters, which in my experience make a real difference to explanation outputs. It's also not clear that using default hyperparameter settings for *all* explanation methods results in a meaningful comparison of what each method is capable of. Settings may need to be tuned to each dataset depending on e.g. number of features.
- It is odd that the synthetic dataset is designed to have no correlations or interactions. This is a good sanity check for post-hoc explanations, but post-hoc explanations are useful only in situations where we have complex correlations/interactions that necessitate a black-box model. It would be more informative to have a different synthetic dataset that has known functions encoding correlations/interactions.

**Additional Feedback:**

N/A

**Correctness:**

The evaluation metrics (rank correlation, stability, etc.) behind the benchmark are appropriate. However, see my comments in Weaknesses about the datasets/models/parameters used in the benchmark.

**Documentation:**

Yes

**Relation To Prior Work:**

Yes

**Summary And Contributions:**

The toolkit satisfies a genuine need for an easy way to evaluate a variety of explanation types over different datasets and models.  This is a step towards being able to evaluate the quality of explanation methods in a more 'objective' way, i.e. by reducing the dependency of such evaluations on the quirks of particular datasets and models.

---

### Official Review · Reviewer_NLvw · 2022-07-27
**Evaluation framework for post-hoc XAI methods**

**Rating:** 8
**Confidence:** 3
**Clarity:** The paper is well-written and accessi…

**Strengths:**

- An urgently needed framework to compare different XAI methods with each other.
- It brings together existing methods, evaluation metrics, and benchmarks under a more comprehensive and user-friendly umbrella
- It has all the important parts of a benchmark/dataset paper: an explanation of usage through API examples, a well-maintained and documented GitHub repo and website; some interesting examples of the ways in which the benchmark paper can be used.
- The paper is well-written

**Weaknesses:**

- The authors write that they "apply FAIR4RS principles and implementation guidelines. However, I don't see this detailed anywhere in the paper. What does it mean? What was done to apply these principles?
- There are existing benchmarks. Perhaps it might make sense to make explicit how the proposed benchmark goes beyond the state of the art (e.g., in terms of API, breadth of metrics, datasets, extensibility, etc.)

**Additional Feedback:**

This is a really great project.

**Correctness:**

The evaluation methods and experiment design are appropriate and performed correctly to the best of my knowledge.

**Documentation:**

The benchmark and datasets of the paper as well as the code are well-documented.

**Ethics:**

XAI methods can be used to provide a false sense of ML methods being "understandable" and "explainable" even though in most cases, these methods cannot provide such a guarantee. The same with fairness measures. In any case, this is not something specific to the paper but in general to the research community working on these methods. And I think that the benchmark provides an important contribution here as it shows the various different metrics and methods and their sometimes contradictory behavior.

**Relation To Prior Work:**

Prior work is fairly mentioned.

**Summary And Contributions:**

The authors propose an evaluation framework for post-hoc explainable AI methods. The framework comes with (i) synthetic data generators as well as real-world datasets, (2) implementations of existing post-hoc XAI methods, (3) a large set of evaluation metrics from the literature to assess post-hoc XAI methods more holistically, and (4) the first ever XAI leaderboard.

---

### Official Review · Reviewer_MX82 · 2022-07-27
**Outdated architectures and algorithms**

**Rating:** 6
**Confidence:** 3
**Clarity:** The paper is very easy to read and un…

**Strengths:**

+Extensible framework for evaluating XAI
+Interesting ideas for synthetic benchmark develoment

**Weaknesses:**

-Mention of Captum is not sufficient.
-No implementation of recent XAI algorithms
-Outdated architectures and models

**Additional Feedback:**

No additional feedback.

**Correctness:**

The submission is correct to the best of this reviewers understanding. The framework makes sense, but the algorithms, modes, and datasets are outdated.

**Documentation:**

Yes, the documentation is appropriate.

**Ethics:**

No concerns.

**Relation To Prior Work:**

The authors need to mention Captum and recent work on XAI.

**Summary And Contributions:**

This paper proposes a framework called OpenXAI for transparent evaluation of model explanations. OpenXAI is comprehensive, extensible, and open source framework for benchmarking post hoc explaination methods. The framework comes with 2 models, 5 datasets, six explainers, and 22 evaluation metrics, and leader boards. The paper has some interesting ideas on designing more fair synthetic data sets (see Section 2.1). Unfortunately,  the remainder of the paper seems to be rather outdated. Integrated gradients is the most resent explanation method that was included in the framework, there has been a substantial amount of work on XAI since then. Many recent methods are integrated into Captum, which is not even mentioned in the related work section. Similar arguments can be made about the baseline modes and datasets.

---

### Official Review · Reviewer_2AP9 · 2022-07-28
**A framework for evaluating post hoc explainers complemented by test datasets and pre-trained models but do not make sufficient novel contributions**

**Rating:** 7
**Confidence:** 5
**Correctness:** The evaluation methodology for the pr…
**Clarity:** The paper is well written and clearly…

**Strengths:**

- The framework provides a simple end-to-end tool for evaluating and benchmarking different models, making the use of explainers more transparent and standardized.
- The framework is easy to use via self-explanatory APIs.
- The framework also provides various datasets, pre-trained models, implemented explainers, and evaluation metrics that make it easy and fast to get started with the tool.


**Weaknesses:**

- The novelty of the work lies only in the compilation of a set of open-source repositories and datasets.
- The framework only includes tabular data and there has been insufficient justification as to why other modalities have not been included.
- Section 3 in on page 6 did not provide enough details on the evaluation metrics and only cited a paper as a reference. It would be beneficial for the reader and the user of the framework to give more details on the mathematical formulations of the metrics. And, for the authors to explain their understanding of these metrics. This applies also to other concepts, models presented in the framework.


**Additional Feedback:**

Since the metrics used to evaluate the explanatory model are an essential part of the framework, the reviewer suggests that a brief explanation of the metrics used be included in the documentation so that it is self-explanatory without having to read the original publications.

**Documentation:**

The paper provides sufficient information on how to use the framework and provides a link to a GitHub repository and web page that provides access to the code and benchmark leaderboard. The methods and metrics used need however more elaboration in the documentation.

**Ethics:**

No concerns were identified.

**Relation To Prior Work:**

The authors have given due consideration to the related work.

**Summary And Contributions:**

The paper presents a framework for evaluating post hoc methods for explaining feature attributes for pre-trained models. The authors supplemented their framework with a collection of 7 datasets (synthetic and real), 2 pre-trained models, 6 explainers, and 22 evaluation metrics. The authors considered three aspects in their framework to standardize the evaluation process of explainers, each aspect includes several evaluation metrics. These aspects are fairness, robustness, and fidelity of explainers.

The main contribution of the paper is the compilation of several open-source projects, datasets, repositories, and the implementation of several evaluation metrics into one tool that makes the evaluation of explainer models more transparent, traceable, reproducible, and easily accessible through various APIs. The authors also developed a novel leaderboard for benchmarking different explanatory methods.

---

### Review · Ethics_Reviewer_E3z8 · 2022-08-22

**Recommendation:** 1

**Ethics Documentation:**

The benchmark presented in this paper assembles a few existing datasets, but the selection rationale and licensing status of each dataset is underspecified. These details should be clarified in the final submission. As the selected datasets are pre-existing, the conditions under which the data were collected remain underspecified in this paper, but the authors might consider adding this information to the OpenXAI website, i.e. a brief description of the data's origin, or links to the original publications to support access to documentation. Regarding the Pima Indians Diabetes Dataset, the authors should consider Radin (2017) '“Digital Natives”: How Medical and Indigenous Histories Matter for Big Data' which discusses the ethical issues concerning the reuse of that dataset.

**Ethics Review:**

The reviewers have flagged some concerns regarding the potentially illusory sense of understanding that may come from the use of explainable AI methods, while also noting that the resources introduced in this paper can support responsible use of these methods. While it is a noteworthy concern about XAI in general, this should not impact the acceptability of this paper.

---

### Author Response · Authors · 2022-08-29
**Summary of Our Discussions and Responses**

Dear Reviewers, AC, and SAC,

Thank you for taking the time to review our work and for engaging in discussions with us. We are very glad to know that all the reviewers appreciate the significance, novelty, and extensiveness of our work. We have had extensive discussions over the past few weeks, and below we summarize these and our responses for the benefit of everyone.

Reviewers mainly had questions about the following two key aspects: i) differences w.r.t. prior XAI libraries and benchmarks, and ii) the rationale behind our choices of XAI methods, datasets, and models.

We addressed the above by not only posting detailed responses below (See the thread on "Response to all reviewers" below, as well as other individual reviewer responses), but also making extensive edits both to the main paper as well as the Appendix. More specifically, in response to reviewer (NLvw, BXQ8, MX82) comments, we included a detailed description of the previously proposed XAI libraries and benchmarks in the main paper (Line 100-158, Section 1), and also described how our work differs from them. In response to reviewer (nA8i) comments, we included two new healthcare datasets (Pima-Indians Diabetes and Framingham heart study) in our framework, and conducted additional experiments to obtain benchmarking results with all the 22 evaluation metrics on these new datasets. Details about these new datasets, additional experiments we conducted, and corresponding results are included in Section E.5 of the Appendix.

Thank you once again for all your time and effort in reviewing this work. We hope we adequately addressed all the questions raised by the reviewers. We will include all the details and clarifications provided as part of our responses in the final version of the paper.

---

### Meta-Review · Area_Chair_QxCR · 2022-09-06

**Recommendation:** Accept
**Confidence:** 4

**Metareview:**

The reviewers unanimously agree that this paper should be accepted.

I would ask the authors to add descriptions of the Pima-Indians Diabetes and the Framingham heart study dataset to Appendix E, and to check the license of the Framingham heart study, as according to Kaggle the lincese is unknown. Furthermore, the readme could be improved by not only describing metrics, but also providing links to the paper giving the full definition.

---

### Decision · Program_Chairs · 2022-09-16

Accept